# Inhibition of TOPORS ubiquitin ligase augments the efficacy of DNA hypomethylating agents through DNMT1 stabilization

DNA hypomethylating agents (HMAs) are used for the treatment of myeloid malignancies, although their therapeutic effects have been unsatisfactory. Here we show that CRISPR-Cas9 screening reveals that knockout of topoisomerase 1-binding arginine/serine-rich protein (*TOPORS*), which encodes a ubiquitin/SUMO E3 ligase, augments the efficacy of HMAs on myeloid leukemic cells with little effect on normal hematopoiesis, suggesting that TOPORS is involved in resistance to HMAs. HMAs are incorporated into the DNA and trap DNA methyltransferase-1 (DNMT1) to form DNA-DNMT1 crosslinks, which undergo SUMOylation, followed by proteasomal degradation. Persistent crosslinking is cytotoxic. The TOPORS RING finger domain, which mediates ubiquitination, is responsible for HMA resistance. In *TOPORS* knockout cells, DNMT1 is stabilized by HMA treatment due to inefficient ubiquitination, resulting in the accumulation of unresolved SUMOylated DNMT1. This indicates that TOPORS ubiquitinates SUMOylated DNMT1, thereby promoting the resolution of DNA-DNMT1 crosslinks. Consistently, the ubiquitination inhibitor, TAK-243, and the SUMOylation inhibitor, TAK-981, show synergistic effects with HMAs through DNMT1 stabilization. Our study provides a novel HMA-based therapeutic strategy that interferes with the resolution of DNA-DNMT1 crosslinks.

DNA hypomethylating agents (HMAs), such as azacitidine (AZA) and decitabine (DAC) are commonly used to treat myeloid malignancies, including myelodysplastic syndrome (MDS) and acute myeloid leukemia (AML)[1]. Upon initial development, they were administered at high doses and patients experienced problematic side effects[2-4]. Later on, AZA was reported to inhibit DNA methyltransferases at much lower concentrations[5]. The current standard doses of DAC, at 20 mg/m²/day, and AZA, at 75 mg/m²/day, are reported to have plasma concentrations at less than 1 μM for most of the post-dose period, as HMAs have a short in vivo half-life[6,7]. Treatment with these low-dose HMAs is generally tolerable and frequently used in elderly patients who are not eligible for intensive chemotherapy[8,9]. However, they are not curative in most cases, and novel therapeutic strategies are required to overcome HMAs[8,10-12].

Although HMAs are widely used in clinical practice, the mechanisms of action remain unclear. HMAs that are incorporated into DNA have a structure similar to cytosine, but the carbon atom to be methylated is replaced by a nitrogen atom, inhibiting DNA methylation[13]. Initially, it was thought that such inhibition of DNA methylation occurred primarily in the promoter regions of tumor suppressor genes, leading to transcriptional activation[14]. However, AML and MDS patients with *TP53* mutations show a better response

✉e-mail: 03aiwama@ims.u-tokyo.ac.jp

to DAC than those with wild-type (WT) *TP53*, a typical tumor suppressor gene, making it difficult to explain the therapeutic effect of HMAs in terms of tumor suppressor gene activation[15]. Recently, it was reported that HMAs incorporated into DNA trap DNA methyltransferase-1 (DNMT1), and form DNA–DNMT1 crosslinks, thereby inducing apoptosis, cell cycle arrest, and mitotic defects[16]. However, the machinery that releases and degrades DNA–DNMT1 crosslinks has not yet been fully characterized.

In this study, we performed a CRISPR-Cas9 knockout (KO) screening, using an sgRNA library that primarily targets epigenetic-related genes, combined with low concentrations of HMAs at less than 1 μM, reflecting the low concentrations that are clinically efficacious, as described previously herein. We have found that KO of topoisomerase 1-binding arginine/serine-rich protein (*TOPORS*), which encodes a ubiquitin/SUMO E3 ligase, enhanced the sensitivity of various MDS/AML cells to HMAs, but had little effect on cell proliferation in the absence of HMAs. Consistently, the deletion of *Topors* in mice augmented sensitivity to DAC in an *MLL-AF9* leukemia model but did not compromise normal hematopoiesis. Furthermore, we demonstrated that DNMT1 was more stable in *TOPORS*-KO cells than in WT cells after DAC exposure due to inefficient ubiquitination. This suggests that TOPORS promotes the degradation of DNA–DNMT1 crosslinks, the accumulation of which is deleterious to cells, as a ubiquitin E3 ligase targeting SUMOylated DNMT1.

## Results

### CRISPR-Cas9 screening reveals that TOPORS KO augments sensitivity to HMAs

To identify molecular targets that enhance the efficacy of HMAs in myeloid malignancies, we performed CRISPR-Cas9 KO screening in the presence of HMAs. We established MDS-L (MDS cell line) and MOLM-13 (MDS/AML cell line) clones (Supplementary Data 1) expressing Cas9, which were then transduced with a lentivirus expressing sgRNA against the *PTPRC* gene encoding CD45 and the *GFP* marker. CD45 expression was downregulated in GFP-positive cells, indicating efficient Cas9-mediated gene editing (Extended Data Fig. 1a, b). These two Cas9-expressing clones were then infected with an sgRNA lentiviral library containing 12,409 sgRNAs targeting 1,383 epigenetic factors, with 2–10 different sgRNAs per gene. To avoid multiple infections, we aimed for a transduction efficiency of less than 30%. Infected cells were selected using puromycin for 48 h, then exposed to low concentrations of HMAs from day 5 post-infection and passed every 72 h for 14 days (Fig. 1a). HMAs were added to the culture with daily refreshments, considering their short half-lives. MDS-L and MOLM-13 cells were screened in the presence of DAC and AZA. The sgRNA profiles were compared between the presence and absence of HMAs, and changes in the sgRNA content for each gene were determined based on β-scores (Fig. 1b–e). In this screening, we focused on sgRNAs, the frequency of which decreased during culture, specifically in the presence of HMAs (β-scores less than −0.3; Supplementary Data 2). Notably, several of these sgRNAs were common in the presence of DAC and AZA (Fig. 1f, g), among which, sgRNAs against *TOPORS*, which encodes a ubiquitin/SUMO E3 ligase (Fig. 1h), was identified in both MDS-L and MOLM-13 cells (Fig. 1f, g, and Extended Data Fig. 2). The sgRNAs against *TOPORS* were repeatedly identified in both cell lines when they were treated with high doses of either DAC or AZA (Extended Data Fig. 3).

### TOPORS-KO MDS/AML cells have enhanced sensitivity to HMAs

To validate the CRISPR-Cas9 screening results, sgRNAs against *TOPORS*, *Cas9*, and *GFP* were introduced into MOLM-13, MDS-L, SKK-1 (MDS/AML cell line), and SKM-1 (MDS/AML cell line) cells (Supplementary Data 1) via lentivirus. Cas9 activity was also confirmed by the downregulation of CD45 expression in SKK-1 and SKM-1 cell lines (Extended Data Fig. 1c, d). Efficient DNA editing of *TOPORS* using two

sgRNAs (sg*TOPORS*#1 and #2) was confirmed in MOLM-13 and MDS-L cells (Extended Data Fig. 4a, b).

Competitive growth assays were performed by co-culturing parental and *TOPORS*-KO cells and evaluated differences in sensitivity to DAC or AZA by monitoring the percentage of GFP-positive *TOPORS*-KO cells (Fig. 2a). In all cell lines, GFP-positive *TOPORS*-KO cells decreased over time in the presence of HMAs, but not in the presence of cytarabine, another cytosine analog (Fig. 2b). Loss of the *TOPORS* gene was assessed in human leukemic cells generated by transforming human cord blood progenitor cells with the *MLL-AF9* leukemia fusion gene[17]. sgRNAs against *TOPORS* and *RFP* were transduced into Cas9-expressing leukemic cells co-cultured with RFP-negative WT cells. The proportion of RFP-positive *TOPORS*-KO cells decreased over time only in the presence of DAC (Fig. 2c), indicating that *TOPORS* KO increased the sensitivity of *MLL-AF9* leukemic cells to DAC. To determine the effect of TOPORS inhibition on normal human hematopoiesis, sgRNA against *TOPORS* was introduced into human CD34-positive cord blood hematopoietic stem and progenitor cells (HSPCs), with a gene editing efficiency of nearly 90%. *TOPORS*-KO HSPC proliferation was not significantly affected (Extended Data Fig. 4e), indicating negligible off-target effects of TOPORS inhibition.

We then established single-cell clones of *TOPORS*-KO MOLM-13 and MDS-L cells using two different sgRNAs (Extended Data Fig. 4c, d) and confirmed *TOPORS* KO in MDS-L cells via western blotting (Fig. 2d). The growth of all *TOPORS*-KO clones was dramatically suppressed only in the presence of DAC (Fig. 2e, f). Sensitivity to DAC was determined by calculating the absolute IC50 from MTS assay data, which confirmed the enhanced sensitivity of *TOPORS*-KO MOLM-13 and MDS-L cells to DAC (Extended Data Fig. 5a). Furthermore, we evaluated protein levels of TOPORS in each cell line (Extended Data Fig. 5b, c) and compared them with the absolute IC50 values to DAC. A positive correlation trend was present between the protein levels of TOPORS and DAC resistance (based on absolute IC50 values), although the correlation was not statistically significant ($p = 0.075$) (Extended Data Fig. 5d).

### TOPORS KO enhances DAC effects in xenograft MDS/AML models

Next, we assessed how KO of *TOPORS* influenced DAC treatment effects in a xenograft mouse model. To precisely monitor tumor burden in mice, we used the AkaBLI system, which is composed of AkaLumine-HCl and Akaluc[18,19]. We transduced MOLM-13 and MDS-L cells with Akaluc using a lentivirus. WT and *TOPORS*-KO MOLM-13 ($5 \times 10^6$) and MDS-L ($1 \times 10^7$) cells were intravenously inoculated into NOG mice and NOG mice expressing human IL-3 and GM-CSF (NOG/IL-3/GM-CSF), respectively, without preconditioning. Because MDS-L is an IL-3-dependent cell line, we used NOG mice expressing human IL-3 and GM-CSF for these xenograft experiments. Recipient mice inoculated with MOLM-13 cells were then treated with 0.3 mg/kg DAC three times per week until recipient mice died, and those with MDS-L cells with 0.3 mg/kg DAC twice per week for 12 weeks (Fig. 2g, h). MOLM-13 cells, which exhibit AML-like features, rapidly expand and induce lethal disease in NOG mice. DAC alone had a mild effect on the survival of recipient mice; however, the combination of DAC and *TOPORS* KO significantly prolonged survival of recipient mice (Fig. 2g). By contrast, MDS-L cells, which retain MDS-like features, expand slowly in mice. We monitored tumor burden using bioluminescence signals in in vivo imaging assays. Eight to 20 weeks after transplantation, the tumor burden from *TOPORS*-KO cells was significantly reduced compared to WT MDS-L cells (Fig. 2h), thereafter, the recipient mice died gradually. Even after cessation of treatment, survival was significantly prolonged in mice transplanted with *TOPORS*-KO cells and treated with DAC (Fig. 2h).

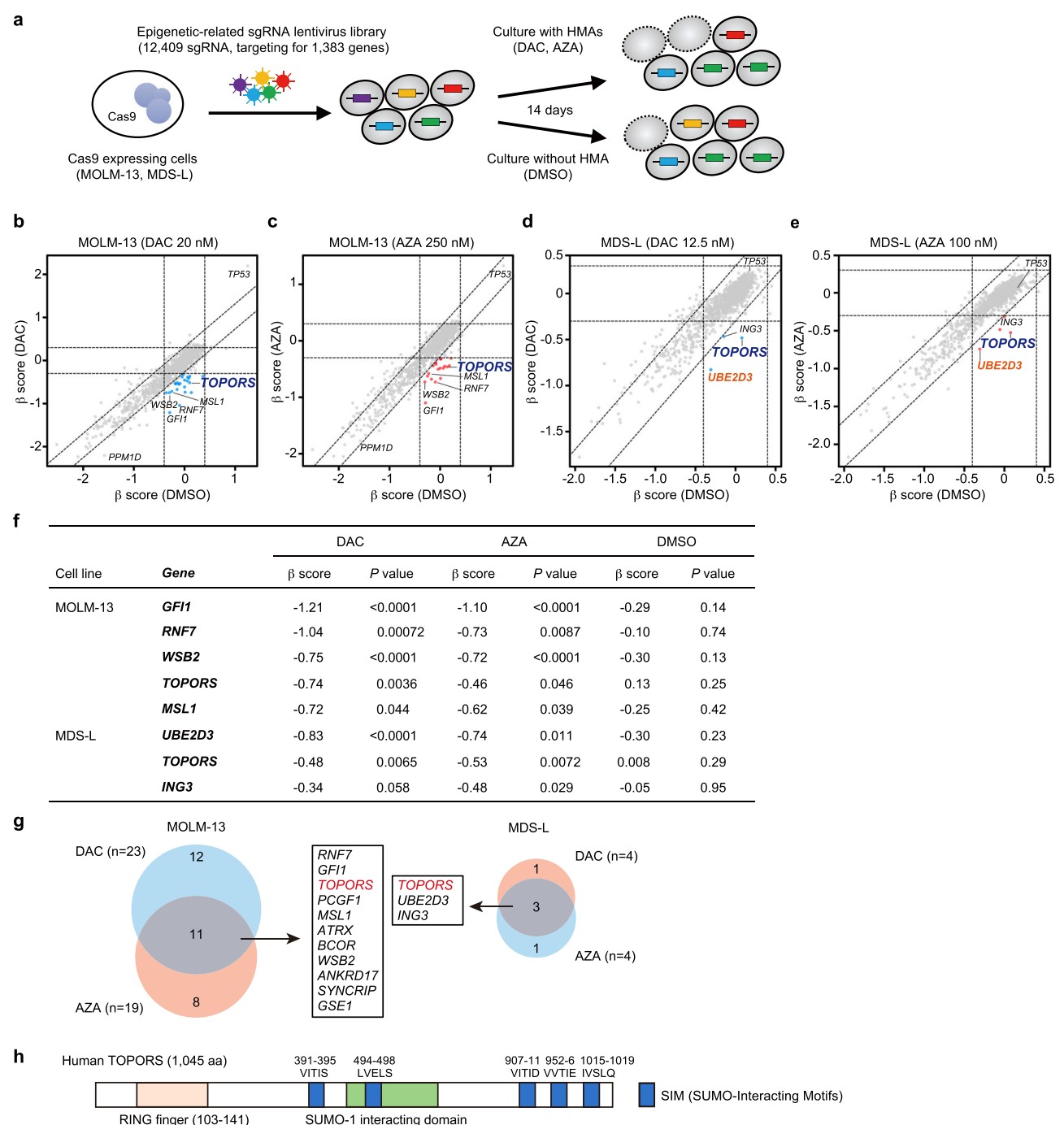

**Fig. 1 | CRISPR-Cas9 screening reveals that *TOPORS* KO augments sensitivity to HMAs. a** Outline of CRISPR-Cas9 screening with and without HMAs. Screening results using MOLM-13 cells treated with DAC at 20 nM, (**b**) and AZA at 250 nM, (**c**) and MDS-L cells treated with DAC at 12.5 nM, (**d**) and AZA at 100 nM (**e**). Scatter plots showing the β scores of each gene, which indicate changes in the sgRNA content for each gene, in the presence and absence of HMAs. Candidate genes whose sgRNAs decreased during culture specifically in the presence of HMAs are indicated by blue dots (DAC) and red dots (AZA). **f** List of candidate genes identified to be in common in the presence of DAC and AZA. The *p*-value was calculated using a two-sided test. **g** Venn diagrams comparing screening results and the list of candidate genes identified to be in common in the presence of DAC and AZA. (**h**) Schematic representation of the TOPORS protein with the RING finger domain, the SUMO-1 interacting domain, and putative SUMO-interaction motifs (SIMs). Source data are provided as a Source Data file.

## Topors is largely dispensable for normal hematopoiesis but its KO sensitizes MLL-AF9 leukemic cells to DAC

To clarify the physiological function of TOPORS, we generated *Topors*-deficient mice in which a large portion of exon 3, including the region encoding amino acids 67−782, was replaced with a neo cassette (Fig. 3a). The *Topors*-deficient genotype was confirmed using PCR (Fig. 3b). *Topors*-deficient mice with a hybrid 129/OlaHsd and C57BL/6 J background showed frequent perinatal lethality and were smaller than their littermates[20]. Correspondingly, *Topors*-deficient mice in a C57BL/6 J background were also recovered at a lower ratio than the expected Mendelian ratio at 4 weeks after birth (WT, 29.6%; *Topors*[+/−], 56.8%; *Topors*[−/−] 13.6%; *n* = 162) but were almost normal in size (data not shown).

Hematopoiesis in WT and *Topors*[−/−] mice (2−4 months old) was then analyzed. No significant changes were observed in the peripheral

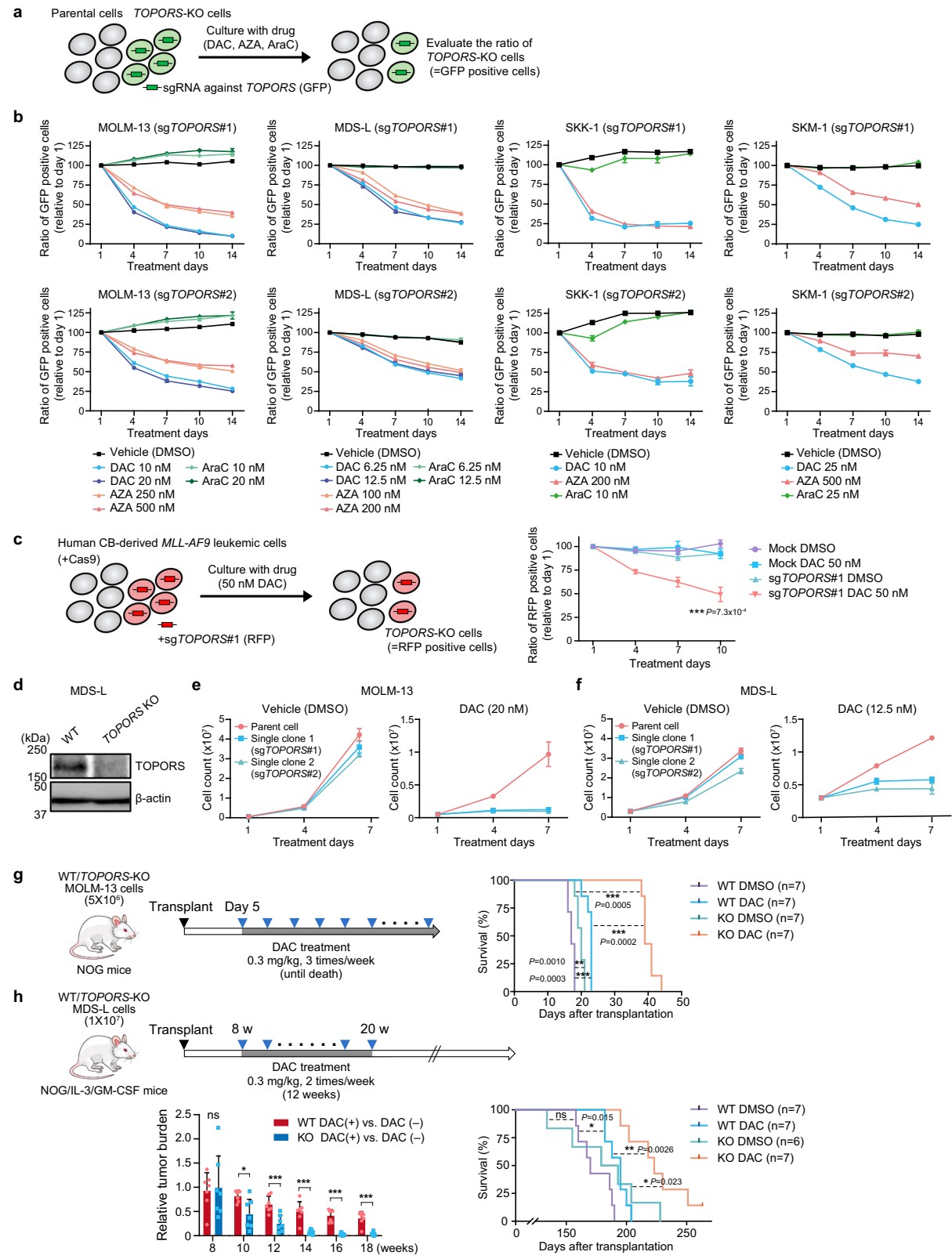

blood (PB; Fig. 3c, d, and Extended Data Fig. 6a) or in the percentage of bone marrow (BM) HSPCs in $Topors^{-/-}$ mice (Fig. 3e and Extended Data Fig. 6b). Next, we transplanted BM cells from WT and $Topors^{-/-}$ mice into lethally irradiated mice. $Topors^{-/-}$ BM cells successfully restored hematopoiesis at levels comparable to those of WT BM cells 4 months after transplantation (Fig. 3f). These data suggest that $Topors$ is dispensable for normal hematopoiesis.

Next, we compared the sensitivity of WT and $Topors^{-/-}$ leukemic cells to HMAs. To this end, we retrovirally transduced WT and $Topors^{-/-}$ granulocyte/macrophage progenitors (GMPs) with $MLL$-$AF9$, seeded them onto semi-solid medium supplemented with SCF, GM-CSF, Flt3L, IL-3, and IL-6, then transferred the cells into liquid culture supplemented with SCF, GM-CSF, IL-3, and IL-6 in the presence of 10 nM DAC. The genotypes of $MLL$-$AF9$ leukemic cells were confirmed with PCR

**Fig. 2 | *TOPORS*-KO MDS/AML cells have an enhanced sensitivity to HMAs.**
**a** Outline of competitive growth assays with HMAs and cytarabine. **b** Proportions of GFP-positive *TOPORS*-KO cells in culture, relative to those at day 1 in competitive growth assays using MOLM-13, MDS-L, SKK-1, and SKM-1 cells (n = 3 for each drug concentration, technical replicates, the experiments were repeated twice independently). Two different sgRNAs against *TOPORS* (sg*TOPORS*#1 and sg*TOPORS*#2) were tested. **c** Proportions of RFP-positive *TOPORS*-KO cells in culture, relative to those at day 1 in competitive growth assays using human *MLL-AF9*-transformed leukemic cells, in the presence and absence of 50 nM DAC (n = 3 for each group, biological replicates). **d** Western blot data of TOPORS in WT and *TOPORS*-KO MDS-L single clones. β-actin served as a loading control. The samples derive from the same experiment but different gels were processed in parallel. Growth of WT and *TOPORS*-KO MOLM-13 (**e**) and MDS-L (**f**) single clones in the presence and absence of DAC (n = 3 for each group, technical replicates, the experiments were repeated twice independently). Evaluation of the sensitivity of WT and *TOPORS*-KO MOLM-13 (**g**) and MDS-L cells (**h**) to DAC in xenograft models using NOG and NOG/IL-3/GM-CSF mice, respectively. Treatment timing and overall survival are depicted. Seven mice were analyzed for each group except for *TOPORS*-KO MDS-L vehicle (n = 6, biological replicates). Tumor burden in MDS-L recipients during DAC treatment (8–20 weeks post-transplantation) was measured by detecting Akaluc bioluminescence signals and relative tumor burden in DAC-treated groups compared with those in non-treated groups is depicted ((**h**), lower left panel). *p < 0.05; **p < 0.01; ***p < 0.001; n.s., not significant by unpaired two-tailed Student's *t*-test. Data are presented as mean ± SD. Source data are provided as a Source Data file.

(Extended Data Fig. 7a). *Topors*$^{-/-}$ leukemic cells showed a similar growth rate to that of WT leukemic cells in the absence of DAC but had significantly inferior growth in the presence of DAC compared to DAC treated WT cells (Fig. 3g). In contrast, DAC exposure did not significantly alter the growth of BM c-Kit$^{+}$ hematopoietic progenitors, indicating a leukemic cell-specific effect by Topors inhibition (Fig. 3h). While the Topors protein was not clearly detected by western blotting, RT-PCR revealed that *Topors* mRNA levels were higher in leukemic cells than in normal c-Kit-positive HSPCs (Extended Data Fig. 7b).

To evaluate the sensitivity of *MLL-AF9* leukemic cells to HMAs in vivo, these cells were transplanted into lethally irradiated recipient mice, with support from CD45.1 mouse BM cells. Recipient mice were treated with DAC for 2 weeks, starting at 4 weeks after transplantation (Fig. 3i). In contrast to WT leukemic cells, which increased in the PB during treatment, *Topors*$^{-/-}$ leukemic cells significantly decreased in proportion and in absolute numbers during treatment. This indicates that *Topors*$^{-/-}$ leukemic cells are more sensitive to DAC than WT leukemic cells (Fig. 3i).

## TOPORS-KO MDS/AML cells show intensified apoptosis and mitotic defects upon DAC exposure

To further understand the enhanced cytotoxic effects of *TOPORS* KO combined with HMAs, we examined apoptosis induction upon DAC treatment and found an increase in Annexin V-positive apoptotic cells among *TOPORS*-KO MDS-L (Fig. 4a) and MOLM-13 (Extended Data Fig. 8a) cells compared to WT cells. Cell cycle analysis revealed that *TOPORS*-KO cells accumulated in the G2/M phase and were reduced in the S phase upon DAC treatment, in both MDS-L (Fig. 4b) and MOLM-13 (Extended Data Fig. 8b) cells. Interestingly, the treatment of MDS-L cells with DAC, in the absence of TOPORS, led to the induction of aneuploidy (Fig. 4b), suggesting that the loss of TOPORS enhances mitotic defects.

Next, we performed RNA sequencing of cells treated with low concentrations of DAC for 48 h. Principal component analysis (PCA) revealed different transcriptomic profiles between the WT and *TOPORS*-KO cells in the presence and absence of DAC (Extended Fig. 8c). We identified differentially expressed genes (DEGs) in *TOPORS*-KO cells treated with DAC, compared to those in WT cells treated with DAC (Supplementary Data 3). This led to the identification of 235 upregulated and 150 downregulated DEGs, and 125 upregulated and 254 downregulated DEGs in MOLM-13 and MDS-L cells, respectively.

## The RING finger domain of TOPORS is responsible for HMA resistance

All data thus far indicate that TOPORS ubiquitin/SUMO E3 ligase antagonizes HMAs and may be involved in HMA resistance. To identify the domain responsible for HMA resistance, we evaluated a deletion mutant of the RING-finger domain, which mediates E3 ubiquitin ligase activity (TOPORS-RD). The addback of WT TOPORS, but not the TOPORS RD mutant, largely abrogated the increased sensitivity of *TOPORS*-KO MDS-L cells to DAC (Fig. 5a). In addition, we created C-to-A mutants at C1 and C1C2 among the conserved cysteine residues in the RING finger domain (Extended Data Fig. 9a). These mutants retain the RING finger domain but show impaired ubiquitination activity[21]. The addback of these mutants was less effective at suppressing DAC resistance than that of WT TOPORS (Extended Data Fig. 9b). The C1 and C1C2 mutants showed intermediate phenotypes between those of the WT and RD mutants, possibly due to incomplete inactivation of catalytic activity. These results further support the hypothesis that the RING finger domain is responsible for DAC resistance. We also evaluated protein levels via western blotting and found that the WT, C1 mutant and C1C2 mutant were hardly detected, while the RD mutant was strongly detected (Extended Data Fig. 9c). Since the RD mutant, which is more stable than WT TOPORS, scarcely suppressed DAC resistance, and the RING finger point mutants were less effective in preventing DAC resistance, it would be reasonable to conclude that the RING finger domain is the responsible site. TOPORS has been reported to function as an E3 ubiquitin ligase with the E2 enzymes UBE2D1, UBE2D3, and UBE2E1[22]. In the first CRISPR-Cas9 screening, we identified *UBE2D3*, a member of the E2 family, as the candidate gene (Fig. 1b). We tested *UBE2D3* KO in MDS-L cells using three different sgRNAs against *UBE2D3* and observed a significantly enhanced sensitivity to HMAs (Extended Data Fig. 9d).

## DNMT1 is stabilized in TOPORS-KO cells

To identify the ubiquitination substrates of TOPORS upon HMA exposure, a second screening was performed using *TOPORS*-KO MOLM-13 and MDS-L single-cell clones and the same sgRNA library targeting epigenetic factors in the presence of DAC (Fig. 5b). We searched for genes that reversed the increased sensitivity of *TOPORS*-KO cells to DAC. Genes with increased β-scores greater than 0.5 for MOLM-13, and greater than 1 for MDS-L, in the presence of DAC compared with in the absence of DAC were extracted as candidates (Fig. 5c, Supplementary Data 4). Twenty-one and 22 genes were extracted from MOLM-13 and MDS-L cells, respectively, but only *UHRF1* and *DNMT1* were identified from both cell lines (Fig. 5d). All sgRNAs targeting *UHRF1* and *DNMT1* in the library decreased the relative percentage of transduced *TOPORS*-KO cells in the absence of DAC but attenuated the reduction or increased the relative percentage of transduced *TOPORS*-KO cells in the presence of DAC (Extended Data Fig. 10a). During DNA replication, UHRF1 specifically binds to hemimethylated DNA and monoubiquitylates multiple lysine residues on histone H3[23]. Multiple monoubiquitylated histone H3 is bound by DNMT1 through two ubiquitin-binding modules in DNMT1-RFTS[24], thereby acting as a platform to recruit and activate DNMT1. Additionally, UHRF1 monoubiquitylates two conserved lysine residues at the N-terminus of PCNA-associated factor 15 (PAF15). Ubiquitinated PAF15 promotes the localization and activation of DNMT1 at DNA methylation sites via specific interactions with DNMT1[25]. These results suggest that the UHRF1–DNMT1 axis is involved in the increased sensitivity of *TOPORS*-KO cells to DAC.

Therefore, we evaluated the protein levels of these factors in MDS-L cells with western blotting. The protein levels of DNMT1, but not UHRF1, were slightly increased in *TOPORS*-KO cells after treatment

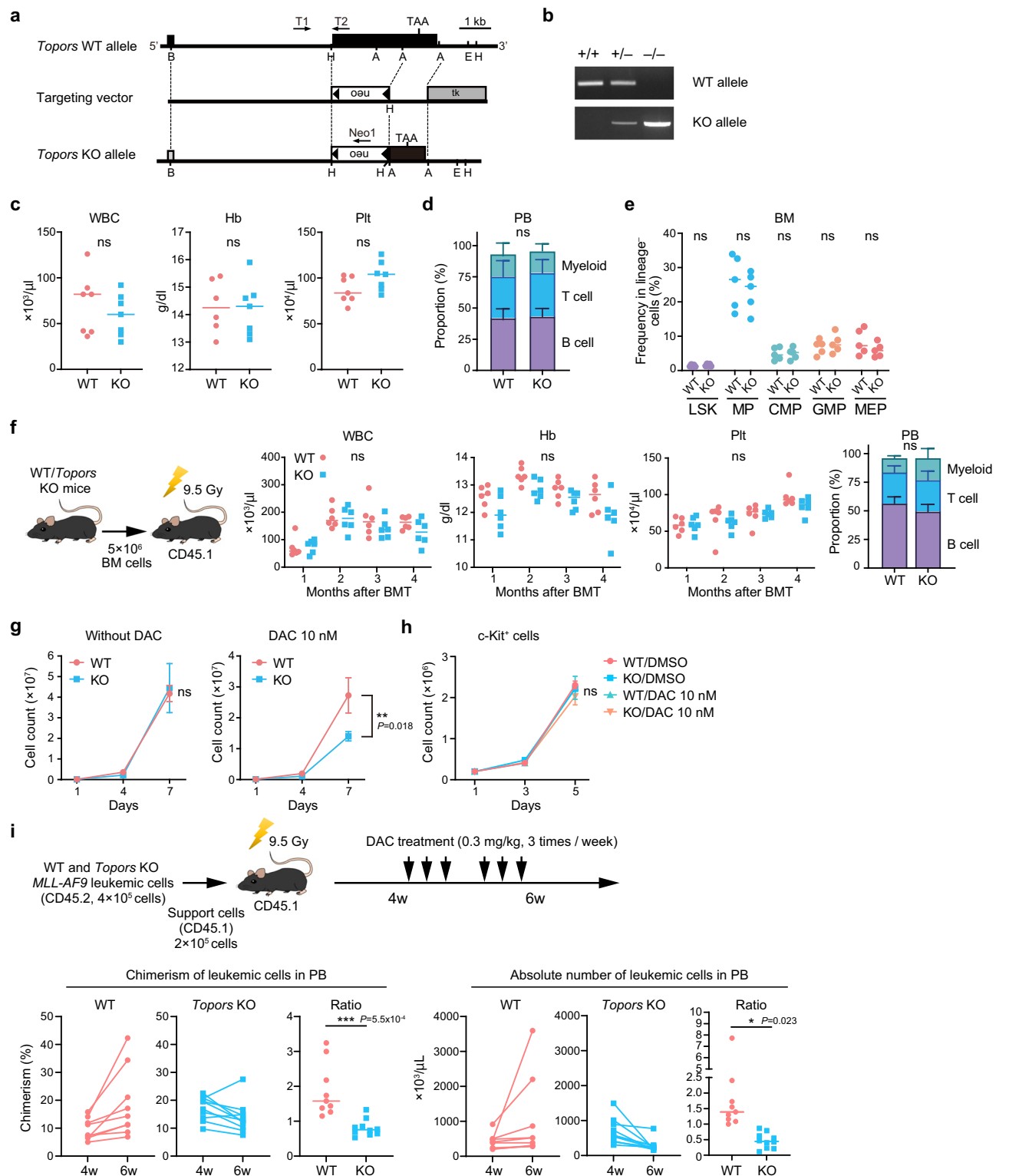

with low-dose DAC (12.5 nM) (Fig. 5e). HMAs are cytosine analogs incorporated into DNA that act as pseudo-substrates for DNMT1, resulting in the covalent entrapment of the enzyme through the formation of a DNA–DNMT1 crosslink. DNMT1 in this crosslink is subsequently degraded[16,26]. DNMT1 degradation after low-dose DAC (12.5 nM) treatment was slightly delayed and alleviated in *TOPORS*-KO cells (Fig. 5e), indicating stabilization of DNMT1 in the absence of TOPORS. Slowly migrating DNMT1 sub-bands, indicating post-translational modifications of DNMT1, were more evident after DAC treatment in *TOPORS*-KO cells (Fig. 5e). To acquire more clear

differences between WT and *TOPORS*-KO cells, western blotting was performed after exposure to high concentrations of DAC (10 μM). DNMT1 stabilization (unmodified main bands) and accumulation of modified DNMT1 (sub-bands) were accentuated upon treatment with a high dose (10 μM) of DAC for 8 h in *TOPORS*-KO MDS-L cells (Fig. 5f, i) as well as *TOPORS*-KO MOLM-13 cells (Fig. 5g, i) and mouse *MLL-AF9* leukemic cells (Fig. 5h, i). DAC-induced degradation of unmodified DNMT1 was evident in WT MDS-L cells and largely attenuated in the presence of the proteasome inhibitor, MG132. In contrast, the effect of MG132 was not obvious in *TOPORS*-KO MDS-L cells, supporting

**Fig. 3 | *Topors* is largely dispensable for normal hematopoiesis. a** Strategy for making a knockout allele for *Topors* by homologous recombination in embryonic stem cells. A large portion of exon 3 encoding amino acids 67 to 782 was replaced by a neo-cassette. **b** Representative PCR-based genotyping of mice obtained from breeding heterozygotes. Tail DNA was used as a template. Complete blood counts (**c**), proportions of myeloid cells (My; Mac-1$^+$ and/or Gr-1+), B220+ B cells, and CD4$^+$ or CD8+ T cells in PB (**d**), and proportions of BM HSPCs (**e**) in WT and *Topors*$^{-/-}$ mice (2–4 months old) (*n* = 5 for each group, biological replicates). LSK Lin$^-$Sca-1$^+$c-Kit$^+$; MP myeloid progenitor; CMP common myeloid progenitor; GMP granulocyte/ macrophage progenitor; MEP megakaryocyte/erythrocyte progenitor. **f** Transplantation assays. Total BM cells (5 × 10$^6$) from WT (red) and *Topors*$^{-/-}$ (blue) mice were transplanted into lethally irradiated CD45.1 recipient mice. Complete blood counts and proportions of myeloid, B, and T cells in PB at 6 months post-transplantation are depicted (*n* = 6, biological replicates). **g** Growth of WT and

*Topors*$^{-/-}$ *MLL-AF9*-transformed cells in the presence and absence of DAC (*n* = 3 for each group, biological replicates). **h** Growth of WT and *Topors*$^{-/-}$ c-Kit-positive cells in the presence and absence of DAC (*n* = 3 for each group, biological replicates). **i** In vivo treatment of *MLL-AF9* leukemic cells with DAC. WT and *Topors*$^{-/-}$ *MLL-AF9* leukemic cells were transplanted into lethally irradiated recipient mice with support BM cells from CD45.1 mice. The recipient mice were treated with DAC from 4 to 6 weeks after transplantation. Chimerism of CD45.2 leukemic cells (left panels) and their absolute numbers (right panels) in PB at 4 and 6 weeks. The values of chimerism and absolute numbers at 6 weeks relative to those at 4 weeks (ratio) are also plotted, respectively (*n* = 9 for WT group, *n* = 10 for *Topors*$^{-/-}$ group, biological replicates). *p < 0.05; **p < 0.01; ***p < 0.001; n.s., not significant by unpaired two-tailed Student's *t*-test. Data are presented as mean ± SD. Source data are provided as a Source Data file.

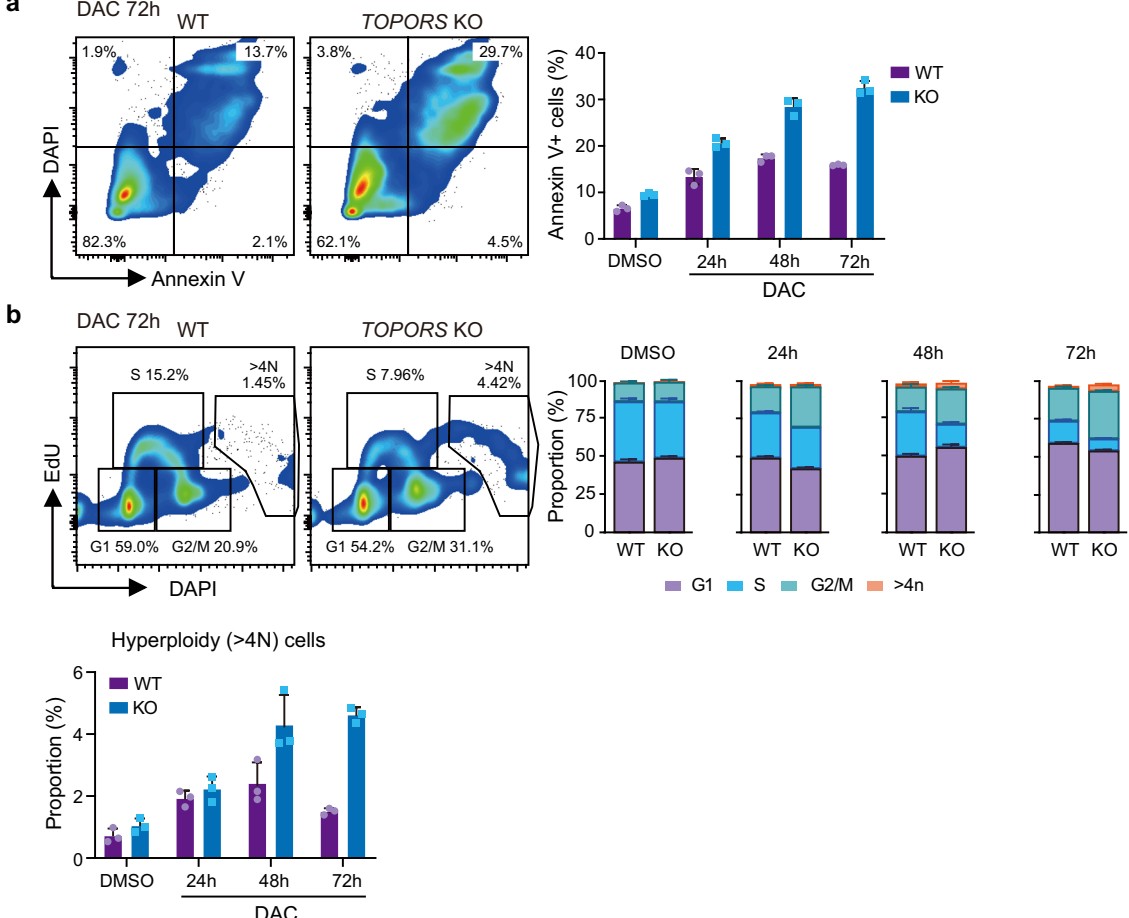

**Fig. 4 | *TOPORS*-KO MDS/AML cells show increased apoptosis and mitotic defects upon DAC treatment. a** Frequency of apoptotic cell death in WT and *TOPORS*-KO MDS-L cells after exposure to 12.5 nM of DAC. Representative flow cytometric profiles of cells at 72 h of DAC exposure (left panel). Percentage of Annexin V-positive cells (*n* = 3 for each group, technical replicates, the experiments were repeated twice independently) (right panel). **b** Cell cycle status of WT and

*TOPORS*-KO MDS-L cells after exposure to 12.5 nM of DAC evaluated by EdU incorporation and DAPI staining. Representative flow cytometric profiles of cells at 72 h of DAC exposure (left panel). Proportion of each cell cycle (middle panel) and hyperploidy cells at the indicated time points (right panel) (*n* = 3 for each group, technical replicates, the experiments were repeated twice independently). Data are presented as mean ± SD. Source data are provided as a Source Data file.

DNMT1 stabilization in the absence of TOPORS (Fig. 5f). DNMT1–DNA crosslink formation triggers prominent SUMOylation of DNMT1, which promotes the resolution of DNMT1–DNA crosslinks and cell fitness upon HMA treatment[16]. DNMT1 sub-bands were largely eliminated in both WT and *TOPORS*-KO MDS-L cells and DNMT1 was stabilized in WT cells after the addition of ML-792, a SUMO E1 ligase inhibitor, indicating that SUMOylation of DNMT1 occurs upon DAC exposure (Fig. 5f). Importantly, SUMOylation of DNMT1 was not affected in *TOPORS*-KO cells, indicating that TOPORS is not involved in the

SUMOylation of DNMT1 but rather in its proteasome-mediated degradation. Immunoprecipitation of DNMT1 8 h after exposure to high concentrations of DAC confirmed the accumulation of SUMOy-lated DNMT1 represented by high-molecular-weight DNMT1 in western blots (Extended Data Fig. 10b). We assessed DNMT1 protein levels after 8 h of exposure to 10 μM DAC using fractionated samples of MDS-L cells. As expected, the slow migration of DNMT1, which represents SUMOylated DNMT1, was more evident in *TOPORS*-KO cells in the chromatin-bound fraction (Extended Data Fig. 10c).

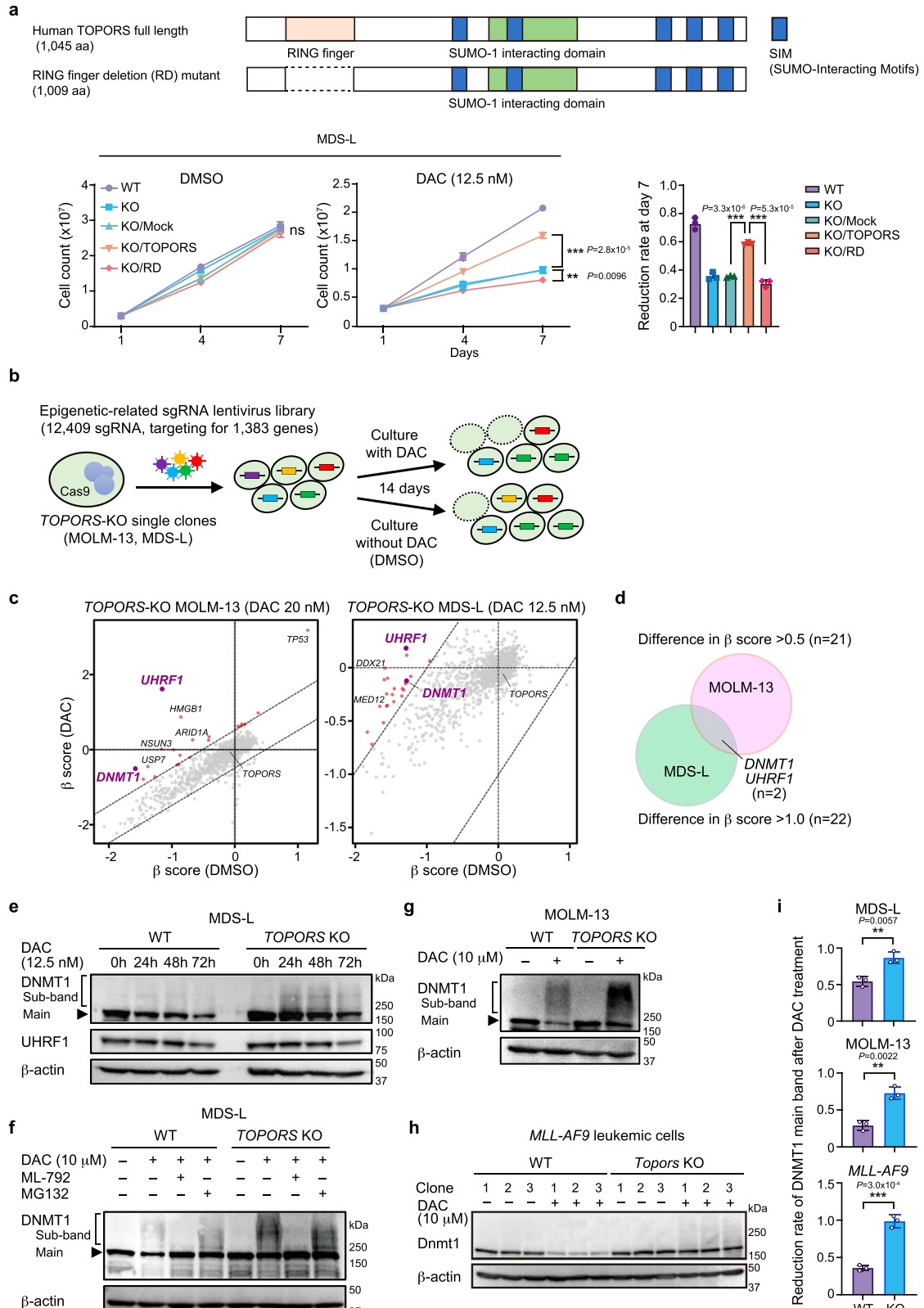

## TOPORS promotes the ubiquitination of SUMOylated DNMT1

RNF4, a SUMO-targeted ubiquitin ligase (STUbL), has been shown to ubiquitinate SUMOylated DNMT1, leading to its proteasomal degradation[27]. However, loss of RNF4 only partially suppressed the removal of SUMOylated DNMT1 from HeLa cells, suggesting the presence of additional STUbLs[27]. Similar to RNF4, TOPORS has a RING finger domain[22] and five putative SUMO-interaction motifs (SIMs) that

recognize SUMOylated substrates when analyzed using the SIM prediction tool, GPS-SUMO (Fig. 1g)[28], therefore, TOPORS may ubiquitinate SUMOylated DNMT1. To address this, mass spectrometry analysis of ubiquitinated proteins in MDS-L cells was performed after exposure to high dose DAC (10 μM, 2 h) (Fig. 6a). Ubiquitinated peptides derived from 328 proteins were more abundant in WT cells than in *TOPORS*-KO cells. Gene Ontology (GO) analysis revealed that proteins associated

**Fig. 5 | DNMT1 is stabilized in *TOPORS*-KO cells. a** Effects of *TOPORS* and *TOPORS*-RD add-back on *TOPORS*-KO MDS-L cell growth in the presence and absence of 12.5 nM DAC (*n* = 3 for each group, biological replicates). Reduction rate of cell numbers on day 7 of culture compared to those of WT MDS-L cells are depicted (right). Full-length *TOPORS* with silent mutations (CGC to AGA) in the sg*TOPORS*#1 target sequence was used. **b** Outline of the second CRISPR-Cas9 screening using *TOPORS*-KO single-cell clones with and without DAC. **c** Scatter plots showing the β-scores of each gene in the presence and absence of DAC. Candidate genes with proportionally increased sgRNA read counts during culture in the presence of DAC compared to those in the absence of DAC are indicated by red and purple dots. **d** Venn diagram showing the two overlapping candidate genes, *UHRF1* and *DNMT1*, between MOLM-13 and MDS-L screenings. **e** Changes in DNMT1 and UHRF1 protein levels after exposure to low-dose DAC (12.5 nM) in WT and *TOPORS*-KO MDS-L cells. Unmodified (main band) and modified (sub-band) DNMT1 are indicated. The samples derive from the same experiment but different gels for DNMT1, β-actin, and another for UHRF1 were processed in parallel. **f** DAC-induced SUMOylation and degradation of DNMT1. WT and *TOPORS*-KO MDS-L cells were exposed to a high dose of DAC (10 μM) for 8 h in the presence and absence of the SUMO inhibitor, ML-792 (3 μM), or the proteasome inhibitor, MG132 (20 μM). Unmodified (main band) and modified (sub-band) DNMT1 are indicated. Changes in DNMT1 protein levels after exposure to high-dose DAC (10 μM) for 8 h in WT and *TOPORS*-KO MOLM-13 cells (**g**) and mouse *MLL-AF9* leukemic cells (**h**). **i** Reduction rate of DNMT1 unmodified main bands after DAC treatment. Intensity of DNMT1 main bands after DAC treatment in (**g**, **h**, and **i**) relative to those before DAC treatment are indicated (*n* = 3 for each group, independent replicates). **p < 0.01; ***p < 0.001; n.s., not significant by unpaired two-tailed Student's *t*-test. Data are presented as mean ± SD. Source data are provided as a Source Data file.

with chromosome organization, DNA repair, mitotic cell cycle, and ubiquitin protein ligase binding were enriched in WT MDS-L cells (Fig. 6b and Supplementary Data 5). DNMT1 was differentially ubiquitinated (Fig. 6c, Supplementary Data 5). We then normalized the abundance of ubiquitinated DNMT1 to the abundance of total DNMT1, as measured by shotgun mass spectrometric analysis of WT and *TOPORS*-KO MDS-L cells (Fig. 6c). All DNMT1-derived peptides were detected more frequently in WT cells than in *TOPORS*-KO cells (Fig. 6c and Supplementary Data 6).

To evaluate the differential degradation speed of DNMT1 with post-translational modifications between WT and *TOPORS*-KO MDS-L cells, 10 μM DAC treatment was applied to both MDS-L cell types for 2 h, the same conditions as the previous mass spectrometric analysis. DNMT1 was immunoprecipitated with an anti-DNMT1 nanobody and subjected to western blot analysis. As expected, SUMOylated DNMT1 accumulated in *TOPORS*-KO cells (Fig. 6d). In addition, the ubiquitination levels of SUMOylated DNMT1 decreased in *TOPORS*-KO cells (Fig. 6d). These data were consistent with the mass spectrometry results (Fig. 6c). DNMT1 was ubiquitinated at lower levels even in the absence of TOPORS (Fig. 6c, d), probably by RNF4. However, considerable accumulation of SUMOylated DNMT1 in *TOPORS*-KO cells upon HMA exposure clearly indicated inefficient proteasomal degradation of DNMT1 in the absence of TOPORS. The ubiquitination of all lysine residues detected in DNMT1 was reduced in *TOPORS*-KO cells compared to WT cells (Supplementary Data 6). Ubiquitination of these residues is likely more dependent on TOPORS than on RNF4 and may promote proteasomal degradation of DNMT1. These results support our findings that TOPORS functions as a SUMO-targeted ubiquitin ligase of SUMOylated DNMT1 in DNA–DNMT1 crosslinks, which undergo proteasomal degradation upon ubiquitination (Fig. 6e). We then checked the initial screening data for RNF4. sgRNAs against RNF4 suppressed cell growth even in the absence of HMAs (Extended Data Fig. 11), suggesting that RNF4 has a broader spectrum of targets than TOPORS as an E3 ligase.

The correlation between *TOPORS* and *DNMT1* expression levels in BM mononuclear cells (MNCs) and AZA treatment response in 23 patients with MDS or AML was investigated (Supplementary Data 7). *TOPORS* expression, but not *DNMT1* expression, was significantly higher in non-responders than in responders (Extended Data Fig. 12). These data support the contribution of TOPORS towards HMA resistance.

### Pharmacological intervention induces mitotic defects via DNMT1 stabilization similar to *TOPORS*-KO background

Finally, we attempted to stabilize DNMT1 to mimic *TOPORS*-KO cells using TAK-243 and TAK-981, small-molecule inhibitors of ubiquitin-activating enzyme (UAE)/E1, and SUMO-activating enzyme subunit 2 (SAE2)[29,30], respectively. As expected, both inhibitors alleviated the degradation of DNMT1 after DAC exposure in WT MDS-L cells, as in *TOPORS*-KO cells (Fig. 7a), and MOLM-13 cells (Extended Data Fig. 13a).

These inhibitors also enhanced growth inhibition mediated by DAC in WT cells, but not in *TOPORS*-KO cells (Fig. 7b and Extended Data Fig. 13b). We also evaluated synergy between HMAs and TAK-243, or TAK-981, in MOLM-13 and MDS-L cells at various concentrations. While combinations were not always synergistic or antagonistic at certain concentrations, those at lower concentrations showed synergistic effects (Extended Data Fig. 13c). Since TOPORS is involved in DNMT1 ubiquitination, we focused on the ubiquitination inhibitor TAK-243 in further experiments. We found an increase in the number of Annexin V-positive apoptotic cells treated with the combination of DAC and TAK-243 (Fig. 7c and Extended Data Fig. 13d). Cell cycle analysis revealed that the combination of DAC and TAK-243 enhanced cell accumulation in the G2/M phase (Fig. 7d and Extended Data Fig. 13e). In a MOLM-13 xenograft model, the combination of DAC and TAK-243 significantly prolonged survival of recipient mice (Fig. 7e). These findings support a therapeutic rationale for targeting the ubiquitination of DNMT1 in DNA-DNMT1 crosslinks induced by HMAs.

### The combination of DAC and TAK-243 is effective against primary AML samples

Two primary BM cells from male patients with AML (Fig. 8a) were cultured in serum-free medium. After the combination treatment of DAC and TAK-243 for 5 days, cell viability was assessed using the MTS assay. Even though each sample showed different responses to the combination, DAC and TAK-243 exhibited synergistic effects (Fig. 8b). Significant reductions in cell viability were observed at representative combinations of concentrations where synergy was particularly strong (Fig. 8b). Next, we transplanted the BM samples from both patients into NOG-W41/IL-3/GM-CSF-immunodeficient mice without pre-conditioning. Only AML cells from patient CH0680 were successfully engrafted and serially transplanted into sub-lethally irradiated NOG/IL-3/GM-SCF mice to establish a patient-derived xenograft (PDX) model (Fig. 8c). At 3 weeks post-transplantation, treatment with DAC and/or TAK-243 began and was administered three times per week while leukemic cell numbers in the PB were monitored. After 2 weeks of treatment, vehicle controls and mice treated with TAK-243 showed a robust increase in leukemic cell numbers in the PB, whereas mice treated with DAC showed a moderate increase. Notably, the combination of DAC and TAK-243 efficiently suppressed leukemia cell expansion and exhibited a significantly enhanced therapeutic effect compared to DAC or TAK-243 alone (Fig. 8d).

### Discussion

The therapeutic efficacy of HMAs for myeloid malignancies is unsatisfactory in most cases, and adjuvant therapies with HMAs have been explored, such as the BCL-2 inhibitor venetoclax[31–35]. However, even with the combination therapy of AZA and venetoclax, long-term remission is rarely achieved. In addition, this combination therapy makes a significant impact on normal hematopoiesis, contributing to frequent side effects, such as severe cytopenia[36]. Therefore, we sought

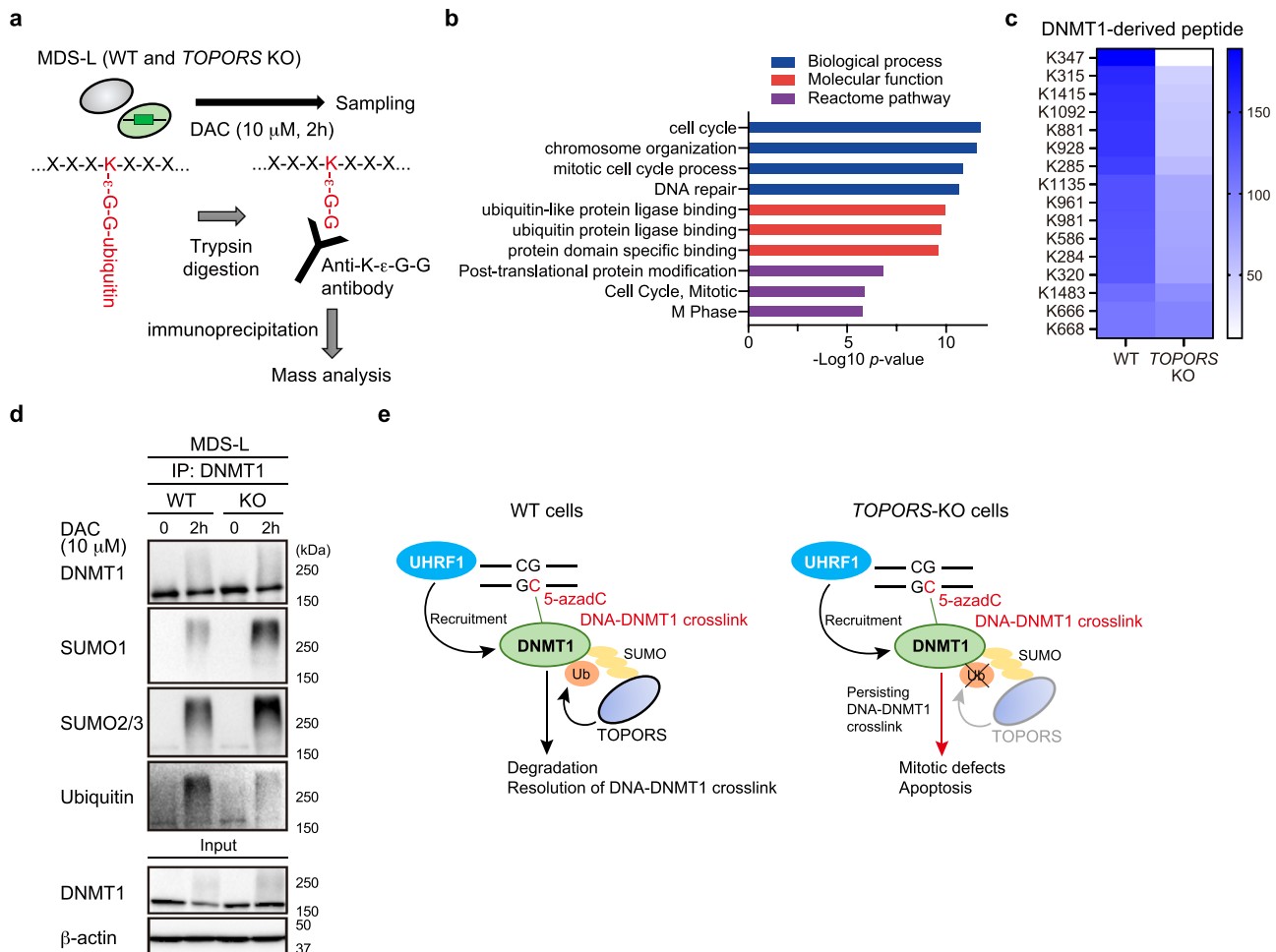

**Fig. 6 | TOPORS promotes ubiquitination of SUMOylated DNMT1. a** Outline of mass spectrometric analysis on ubiquitinated proteins in MDS-L cells after DAC exposure. **b** GO terms enriched in ubiquitinated peptides detected in WT MDS-L cells compared to those in *TOPORS*-KO MDS-L cells. **c** Heat map showing enrichment of the DNMT1-derived ubiquitinated peptides in WT MDS-L cells compared to *TOPORS*-KO MDS-L cells. The abundance of ubiquitinated DNMT1 was normalized to the abundance of total DNMT1. **d** SUMOylation and ubiquitination levels of DNMT1 in WT and *TOPORS*-KO MDS-L cells treated with 10 nM DAC for 2 h. Endogenous DNMT1 immunoprecipitated with anti-DNMT1 nanobody was subjected to western blotting. β-actin was served as a loading control of the inputs. The samples derive from the same experiment but different gels for all were processed in parallel. **e** A model of TOPORS function found in this study. Source data are provided as a Source Data file.

out novel therapeutic target genes that augment sensitivity to HMAs with little effects on normal hematopoiesis. Via CRISPR-Cas9 screening, the KO of *TOPORS* was found to enhance the efficacy of HMAs. This synergistic effect was confirmed in MDS/AML cell lines, *Topors*-deficient GMPs transformed with *MLL-AF9*, and xenograft models. *TOPORS* KO had little effect on cell proliferation in the absence of HMAs. Consistently, deletion of *Topors* in mice had no significant effect on hematopoiesis. These results suggest that the functional inhibition of TOPORS could be a promising therapeutic strategy to increase the therapeutic efficacy of HMAs, with little effect on normal hematopoiesis.

*TOPORS*-KO cells showed increased apoptosis and mitotic defects following HMA treatment. Addback experiments revealed that the site responsible for HMA resistance is the RING finger domain of TOPORS, which is required for ubiquitination. These results strongly suggest that the ubiquitin ligase function of TOPORS is essential for HMA resistance. Although TOPORS has been reported to act as a ubiquitin ligase for p53[22], *TOPORS* KO had the same effect on MDS-L and SKM-1 cells, with biallelic loss-of-function mutations in *TP53*[37,38], as on MOLM-13 and SKK-1 cells with WT *TP53*, suggesting a p53-independent mechanism that augments HMA efficacy. The second screening, using *TOPORS*-KO cells, successfully revealed that both *UHRF1* and *DNMT1*

KO significantly mitigated *TOPORS* KO-enhanced sensitivity to HMAs, providing them as candidate ubiquitination substrates for TOPORS. When phosphorylated, HMAs are incorporated into DNA and trap DNMT1 to form DNA–protein crosslinks (DPCs), causing mitotic defects and cell growth inhibition[16,27]. DNMT1 in DPCs is SUMOylated[16,27], followed by ubiquitination by RNF4, and undergoes proteasomal degradation, resolving the DNA–DNMT1 adducts[27]. Based on these findings, we hypothesized that TOPORS, which has an RNF4-like domain composition matching that of STUbLs, contributes to the degradation of DNA–DNMT1 crosslinks. González-Prieto et al. demonstrated that TOPORS binds with a high affinity to both SUMO1 and SUMO2, which is consistent with this study's hypothesis that TOPORS acts as a STUbL[39]. Similar to the effect of *RNF4* knockdown in HeLa cells[27], DNMT1 was stabilized in *TOPORS*-KO cells, and the resolution of SUMOylated DNMT1 was considerably delayed after DAC exposure. Furthermore, immunoprecipitation of SUMOylated DNMT1 and mass spectrometric analysis of the ubiquitinated peptides indicated that TOPORS acts as a ubiquitin ligase for SUMOylated DNMT1, thus providing new insights into the mechanisms underlying the effects of HMAs. Our screening results also indicate that TOPORS is a better therapeutic target than RNF4, which is required for cell growth, even in the absence of HMAs.

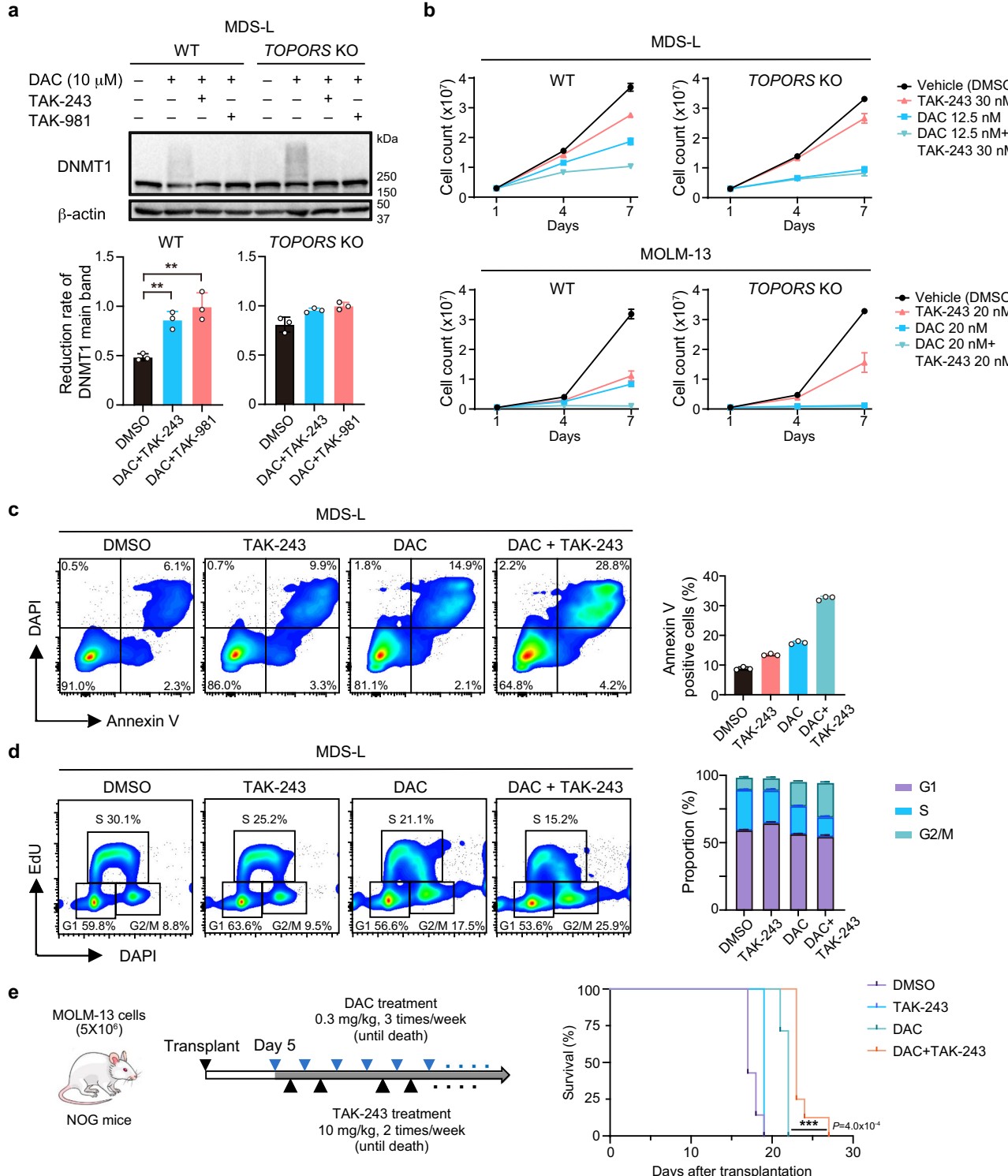

**Fig. 7 | Pharmacological intervention induces mitotic defects via DNMT1 stabilization. a** Changes to DNMT1 protein levels after exposure to DAC in WT and *TOPORS*-KO MDS-L cells. The cells were exposed to a high dose of DAC (10 μM) for 8 h in the presence and absence of TAK-243 (1 μM), or TAK-981 (3 μM) (*n* = 3 for each group, independent replicates). **b** Growth of WT and *TOPORS*-KO MDS-L clones in the presence and absence of DAC and/or TAK-243 (*n* = 3 for each group, technical replicates, the experiments were repeated twice independently). **c** Frequency of apoptotic cell death in MDS-L cells after exposure to 12.5 nM of DAC. Representative flow cytometric profiles of cells at 72 h of DAC exposure (left panel). Percentage of Annexin V-positive cells (*n* = 3 for each group, technical replicates, the experiments were repeated twice independently) (right panel). **d** Cell cycle status of WT

and *TOPORS*-KO MDS-L cells 72 h after exposure to DAC (12.5 nM) and/or TAK-243 (30 nM). Representative flow cytometric profiles of cells (left panel). Proportion of each cell cycle evaluated by EdU incorporation and DAPI staining (*n* = 3 for each group, technical replicates, the experiments were repeated twice independently) (right panel). **e** In vivo treatment of MOLM-13 cells with the combination of DAC and TAK-243. WT MOLM-13 cells were transplanted into NOG mice. The recipient mice were treated with DAC and/or TAK-243 until they died. Treatment timing and overall survival are depicted (*n* = 7 for each group, biological replicates). **p < 0.01; ***p < 0.001; n.s., not significant by unpaired two-tailed Student's *t*-test. Data are presented as mean ± SD. Source data are provided as a Source Data file.

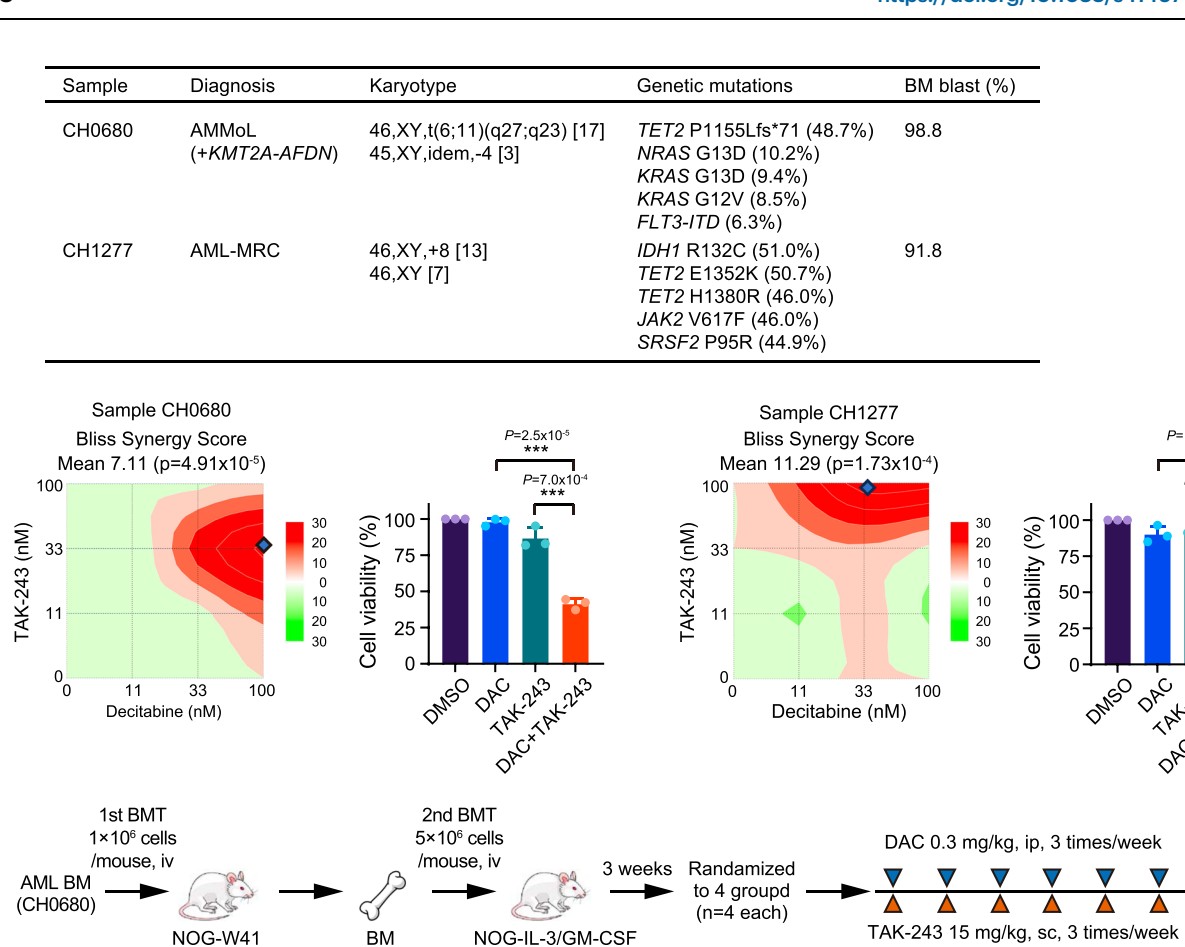

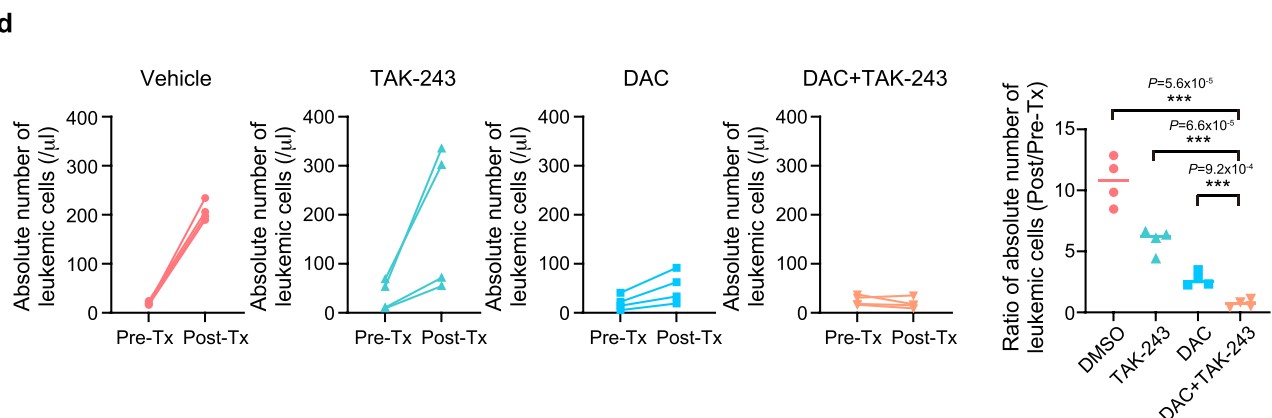

**Fig. 8 | The combination of DAC and TAK-243 is effective for primary AML samples. a** Clinical information and genetic mutations of patient samples. AMMoL, acute myelomonocytic leukemia; AML-MRC, AML with myelodysplasia-related changes. **b** Synergistic effects between DAC and TAK-243 in patients' samples at various combinations of concentrations. Synergy was calculated using the Syner-gyFinder 3.0, and BLISS was used to denote scores. Representative combinations of concentrations at which the synergistic effect was particularly strong are shown by dots. Representative synergistic effects of combinations are also depicted (right panel) (CH0680, DAC 100 nM and TAK-243 33 nM; CH1277, DAC 33 nM and TAK-243 100 nM) (*n* = 3 for each group, biological replicates). **c** Schema of the PDX study. Treatment timing is depicted. **d** Absolute number of human CD45-positive cells in PB and their ratio before and after treatment in each group (*n* = 4 for each group, biological replicates). ***p < 0.001 by the Student's *t*-test. Data are presented as mean ± SD. Source data are provided as a Source Data file.

Preventing the timely resolution of DNMT1 DPCs induced by HMAs causes the accumulation of cells in the G2/M stage and markedly perturbs chromosome alignment at the metaphase plate, resulting in defective chromosome segregation, leading to aneuploidy[27]. The significant accumulation of aneuploid *TOPORS*-KO cells after DAC treatment supports the inefficient resolution of DNMT1 DPCs in the absence of TOPORS. We found that the combination of DAC and the ubiquitination inhibitor, TAK-243, enhanced the effect of DAC on

MDS/AML cells, including primary patient samples, mimicking features observed in *TOPORS*-KO cells. It was also found that DNMT1 was stabilized by the synergistic effects of DAC and TAK-981. Recently, the synergistic effects of HMAs and TAK-981 on hematological malignancies have been reported[40,41]. The current study provides further mechanistic insights into their synergism and proposes that TOPORS is implicated in this process as a STUbL for DNMT1. Taken together, these results indicate that the DNMT1 regulatory pathway, from the

recruitment of DNMT1 to the resolution of DNMT1 DPCs, is a novel target for augmenting the efficacy of HMA treatment.

### Limitations of the study

This study had several limitations. First, while the 2-week in vitro screening identified several genes that may contribute to HMA resistance, the effect of HMAs is often observed several months after treatment initiation. The candidate genes identified may not necessarily overlap with those involved in human treatment susceptibility. However, *TOPORS* KO sensitized MDS-L cells to DAC in xenograft models and prolonged survival after 3 months of DAC treatment. These findings suggest that TOPORS is a promising therapeutic target in clinical practice. The second limitation in this study refers to the results showing TOPORS involvement in the ubiquitination of SUMOylated DNMT1 and contributing to HMA resistance. However, it was difficult to prove that SUMOylated DNMT1 was truly cross-linked to DNA after DAC exposure, although previous studies have reported this connection[16]. Lastly, the ubiquitination inhibitor, TAK-243, had a synergistic effect on primary AML samples through DNMT1 stabilization. Since TAK-243 has various targets, other than TOPORS, strategies to specifically target TOPORS using other modalities, such as oligonucleotide therapeutics[42], or proteolysis-targeting chimera (PROTAC)[43], should be considered for clinical application.

## Methods

### Ethics

All animal experiments were performed in accordance with the Institutional Guidelines for the Use of Laboratory Animals and were approved by the Review Board for Animal Experiments of the University of Tokyo (approval ID: PA18-42). In the experiments using disease models in mice, the tumor burden did not exceed the maximal limits permitted by our institutional review board.

### Cell lines and cell culture

MOLM-13 cells[44] were cultured in RPMI-1640 (Sigma) supplemented with 10% fetal bovine serum (FBS; Corning) and 1% penicillin streptomycin sulfate (PSG; Wako). MDS-L cells were provided by Dr. Kaoru Tohyama (Kawasaki Medical School)[38] and cultured in RPMI-1640 supplemented with 10% FBS and 1% PSG in the presence of 15 ng/mL of recombinant human interleukin-3 (IL-3; BioLegend). SKK-1 cells[45] were cultured in RPMI-1640 supplemented with 10% FBS and 1% PSG in the presence of 10 ng/mL of recombinant human granulocyte-macrophage colony-stimulating factor (GM-CSF; BioLegend). SKM-1 cells[46] were cultured in RPMI-1640 supplemented with 20% FBS and 1% PSG. All cell lines were cultured at 37 °C with 5% $CO_2$. HL-60, THP-1, K562, and HeLa cells were cultured in RPMI-1640 supplemented with 10% FBS and 1% PSG. HEK293T cells were cultured in DMEM (Sigma) with 10% FBS and 1% PSG at 37 °C with 10% $CO_2$. The characteristics of MOLM-13, MDS-L, SKK-1, and SKM-1 cells are listed in Supplementary Data 1. MTS assays were performed according to the manufacturer's instructions (Promega G3580). Synergy effects were calculated using Synergy Finder 3.0[47], and BLISS was used to denote the scores.

### Mice

All animal experiments were performed in accordance with the Institutional Guidelines for the Use of Laboratory Animals and were approved by the Review Board for Animal Experiments of the University of Tokyo (approval ID: PA18-42). Housing condition were temperature $22 \pm 2$ °C, humidity $55 \pm 5$%, light/dark cycle 12 h/12 h (8 a.m.−20 p.m. light). NOD.Cg-Prkdc[scid] Il2rg[tm1Sug]/ShiJic (NOG), NOD.Cg-Prkdc[scid] Il2rg[tm1Sug] Tg(SRa-IL3, CSF2)7-2/Jic (NOG-EXL) mice expressing human IL-3 and granulocyte-macrophage colony-stimulating factor (GM-CSF; NOG/IL-3/GM-CSF)[48], and NOG-EXL mice with the W41 mutation (NOD.Cg-Prkdc[scid] Il2rg[tm1Sug] Kit[em1(V831M)Jic] Tg(SRa-IL3, SRa-CSF2)/Jic) (NOG-W41-EXL) were purchased from In-Vivo Science

Inc. and the Central Institute for Experimental Animals (Kawasaki, Japan), respectively. C57BL/6J mice (Ly5.1) used as recipients for transplantation experiments were purchased from Sankyo Laboratory Services. *Topors*[+/−] mice were generated in this study and were back-crossed to a C57BL/6 J background for over five generations. *Topors*[+/−] mice were crossed to generate *Topors*[−/−] mice. Littermates were used as wild-type (WT) controls. Genotypes were confirmed by PCR, using genomic DNA as a template. The primers used were as follows: TA-Neo1, 5′-GATGGATTGCACGCAGGTTCGC-3′; *Topors*, 5′-primer 5′-GGTCATGACATGAAGTAGCAGGC-#7; *Topors* 3′-primer 5′-GCAGCTGAGGCCATTATCTGGC-#7

### Quantitative RT-PCR

Total RNA was extracted using the RNeasy Micro Plus Kit (QIAGEN) and reverse-transcribed using the SuperScript IV First-Strand Synthesis System (Invitrogen) with an oligo-dT primer. Real-time quantitative PCR was performed using the StepOnePlus Real-Time PCR System (Life Technologies) with TB Green Premix Ex Taq II (TaKaRa Bio). All data are presented as relative expression levels normalized to β-actin expression. The primer sequences used were as follows:

*Topors*, forward 5′-CCGAATGCCCACTGTGTAAAC-3′; reverse 5′-TAGTTGTCCGGTAGCGAAACC-3′; *β-actin*, forward 5′-CTGGCTCCTAGCACCATGAAGATC-3′, reverse 5′-TGCTGATCCACATCTGCTGG-3′

### Plasmids

LentiCas9-Blast (Addgene #52962) was used to transduce *Streptococcus pyogenes* Cas9 and blasticidin resistance genes into each cell line. For screening, a domain-focused lentiviral sgRNA library consisting of 12,409 sgRNAs targeting 1,383 genes mainly related to epigenetic factors was used. The library also contained 1000 negative control sgRNAs and approximately 100 positive control sgRNAs targeting known essential genes. Plasmids were electroporated (0.1 cm gap, 1.8 kV) into ElectroMAX™ Stbl4™ Competent Cells (Thermo Fisher) using the GenePulser Xcell™ (Bio-Rad), which were transferred to SOC Outgrowth Medium (New England BioLabs) for 1 h at 37 °C and seeded onto LB plates with ampicillin. All colonies were collected, and plasmids were extracted and purified using the Genopure Plasmid Maxi Kit (Roche) and used for virus production. For CRISPR-Cas9-mediated gene knockout, the vector LentiCRISPR v2 (Addgene #52961) in the FG12 (Addgene #14884) backbone was used. The vector was provided by Dr. Masaya Ueno (Kanazawa University, Japan). The sgRNA sequences for each gene were incorporated into the BsmBI site of LentiCRISPR v2. The following primer sequences were used: sg*PTPRC*, 5′-GAAACTTGCTGAACACCCGC-3′; sg*TOPORS*#1, 5′-GTTTTCGCTGTGTACAGGAG-#7; sg*TOPORS*#2, 5′-CAGCTGAAATATCCCTGTAG-#7; sg*UBE2D3*#1, 5′-ACTTACAGGTCCCATAATTG-3′; sg*UBE2D3*#2, 5′-GGGAAAATACTTGCCTTAGG-3′; and sg*UBE2D3*#3, 5′-AATGACAGCCCATATCAAGG-3′.

Plasmids containing full-length human *TOPORS* (pCATCH-FLAG-TOPORS) and the RING-finger deletion mutant (pCATCH-FLAG-TOPORS-RD) were provided by Dr. Stefan Wegar (Free University of Berlin, Germany). FLAG-TOPORS, FLAG-TOPORS-RD, FLAG-TOPORS with silent mutations (CGC to AGA) in the sg*TOPORS*#1 target sequence, and the TOPORS-RD C1 mutant (TGT to GCT) and C1C2 mutant (TGC to GCC) were subcloned into the pMYs-IRES-Puro retroviral vector. Prime STAR Mutagenesis Basal Kit (Takara) was used to induce mutations.

### Virus production

For lentivirus production, the pMD2G-VSV-G envelope-expressing plasmid (#12259; Addgene), psPAX2 packaging plasmid (#12260; Addgene), and lentiviral vectors were transfected into HEK293T cells using polyethylenimine. The culture supernatant was replaced once at 12−16 h after transfection, then collected 48−60 h later and filtered through a 0.45 μm filter. For retroviral production, the RD114 envelope

plasmid, M57 gag-pol plasmid, and viral vectors were transfected into 293 T cells using the calcium phosphate method. Alternatively, 293gp cells were used in combination with the pcDNA-VSV-G envelope-expressing plasmid, and the culture supernatant was collected and filtered through a 0.45 μm filter.

## CRISPR-Cas9 screening
The library vectors contained *GFP* and puromycin resistance genes. Infection efficiency was evaluated by monitoring GFP expression. To avoid multiple infections, the infection efficiency was set to less than 30%. Forty-eight hours after infection, 1.5 μg/mL puromycin was added, and drug selection was performed for 72 h to select infected cells. Exposure to HMAs was initiated after the enrichment of GFP-positive cells was confirmed. HMAs were added every 24 h to achieve the stated concentrations, and the cells were passaged once every 3 days and cultured for 14 days. During the incubation period, the average cell count was maintained at 500 cells/sgRNA. PCR amplification was performed using Ex Taq polymerase (Takara) with genomic DNA extracted from the cells as a template. The primer sequences for the amplification of the sgRNA portions and sequencing by next-generation sequencing are listed in Supplementary Data 8. The amplified DNA was purified using AMPure XP (Beckman Coulter #A63880), quantified using TapeStation (Agilent), and sequenced using Hiseq2500 (Illumina). For the obtained FASTQ files, sequences other than the 20-mer sgRNA sequences were trimmed, and sgRNA sequences were extracted using Cutadapt v1.11 and aligned to the sgRNA–gene reference file. The results were analyzed using MAGeCK software 0.5.9.2[49], and changes in the sgRNA content for each gene were indicated by the β-score.

## In vitro competitive growth assays
Cell lines expressing Cas9 were transduced with sgRNA against *TOPORS*, using *GFP* as the marker gene. HMA exposure was conducted using conditions under which 30–50% of *TOPORS*-KO cells coexisted with parental cells and HMAs were refreshed every 24 h. The percentage of GFP-positive cells was evaluated every three days to assess the difference in drug sensitivity between the parental and *TOPORS*-KO cells.

## Apoptosis and cell cycle assays
Apoptosis and the cell cycle were examined using the Annexin V Apoptosis kit (BD Biosciences) and the Click-iT™ Plus EdU Alexa Fluor™ 647 Flow Cytometry Assay Kit (Thermo Fisher Scientific) according to the manufacturer's instructions.

## RNA sequencing
RNA was extracted from cells using the RNeasy Mini Kit (Qiagen). RNA (500 ng) was used to create a library for sequencing using a Next Ultra DNA Library Prep Kit (New England BioLabs). After confirming the concentration of each sample on a TapeStation (Agilent Technologies), deep sequencing was performed using a Hiseq2500 (Illumina). TopHat (version 2.0.13; default parameters) was used to map reads to the reference genome (UCSC/mm10). Gene-level counts of specifically mapped fragments were extracted from BAM files. Gene expression values were calculated using Cufflinks (version 2.2.1) as the number of reads per exon per million mapped reads. Principal component analysis (PCA) of each sample was performed using iDEP 96[50], and gene set enrichment analysis (GSEA) was performed based on data curated by MSigDB, the Broad Institute molecular signature database.

## Xenotransplantation
NOG mice (10 weeks old, female) and NOG/IL-3/GM-CSF mice (10 weeks old, male) were used as MOLM-13 and MDS-L xenografts, respectively. Akaluc was introduced into each cell line using the mScarlet-P2A-Akaluc lentivirus to confirm engraftment after transplantation. One hundred microliters of 5 mM AkaLumine-HCl (Wako) was injected intraperitoneally into the mice immediately before image analysis, and mice under isoflurane anesthesia were imaged within 5–10 min after injection. Signal intensity was measured using an IVIS instrument (Perkin Elmer). Mice with confirmed engraftment were randomly divided into two groups and treated with either DAC and/or TAK-243 or vehicle. The animals with MOLM-13 were monitored daily until all mice died. The animals with MDS-L were euthanized at 9 months post-transplantation.

## Flow cytometric analysis and BM transplantation
To evaluate mouse peripheral blood (PB) and bone marrow (BM) cells, FACSAria III (BD Biosciences) and FACSCelesta (BD Biosciences) were used to evaluate PB and BM cells. Mouse PB cells were collected from the orbital venous plexus using capillaries and erythrocytes were lysed using erythrocyte lysis buffer (150 mM $NH_4Cl$) before analysis. BM cells were extracted from the femur, tibia, and pelvis of the euthanized mice. After erythrocyte lysis, BM cells were filtered through a 45 μm filter and centrifuged on a Histopaque-1119 (Sigma) to separate mononuclear cells. PBS containing 2% FBS was used for cell extraction and antibody suspension. The monoclonal antibodies recognizing the following antigens were used in flow cytometry and cell sorting: Gr1-e450 (RB6-8C5, Invitrogen), Mac1-APC (M1/70, BioLegend), Ter119-PECy5 (TER-119, BioLegend), B220-A700 (RA3-6B2, BioLegend), CD3-PECy7 (145-2C11, BioLegend), CD45.1-FITC (104, BioLegend), CD45.2-APC/Cy7 (A20, BioLegend). The following antibodies were used for the immature fraction; CD3e-PECy5 (145-2C11, eBioscience), CD4-PECy5 (GK1.5, BioLegend), CD5-PECy5 (53-7.3, BioLegend), CD8a-PECy5 (53-6.7, BioLegend), B220-PECy5 (B298291, BioLegend), Mac1-PECy5 (M1/70, BioLegend), Gr1-PECy5 (RB6-8C5, BioLegend), Ter119-PECy5 (TER-119, BioLegend), cKit-APC (2B8, BioLegend), ScaI-PECy7 (D7, BioLegend), CD34-FITC (RAM34, invitrogen), FcgR-BV510 (93, BioLegend). Lineage cells in PB were defined as Mac-1- and/or Gr-1-positive for the myeloid lineage, CD19-positive for the B-cell lineage, or CD3-positive for the T-cell lineage. Hematopoietic stem and progenitor cells in BM were defined as lineage-negative, Sca-1-positive, c-Kit-positive for LSK, lineage-negative, Sca-1-negative, c-Kit-positive for myeloid progenitors, and among myeloid progenitors, CD34-positive and FcγRII/III-negative for common myeloid progenitors (CMPs), CD34-negative, FcγRII/III-negative for megakaryocyte-erythrocyte progenitors (MEPs), and CD34-positive, FcγRII/III-positive for granulocyte-macrophage progenitors (GMPs). Flow cytometry data were analyzed using FlowJo v10.6.1.

Mononuclear cells were isolated from the BM of WT and *Topors*[-/-] mice and transplanted via the tail vein into lethally irradiated (9.5 Gy) recipient mice. PB was analyzed monthly after transplantation, and PB and BM were analyzed 4 months after transplantation.

## *MLL-AF9* leukemogenesis
GMPs isolated from WT or *Topors*[-/-] male mice were cultured in SF-O3 (Sanko Junyaku) supplemented with 0.2% BSA, 1% PSG, murine stem cell factor (SCF; 40 ng/mL), GM-CSF (20 ng/mL), Flt3L (20 ng/mL), IL-3 (20 ng/mL), and IL-6 (20 ng/mL) for 24 h and then transduced with the *MLL-AF9* retrovirus using Retronectin (Takara). After 48 h of infection, the cells were transferred to methylcellulose medium (MethoClut™) (STEMCELL Technologies, M3234) supplemented with SCF (50 ng/mL), GM-CSF (10 ng/mL), IL-3 (10 ng/mL), and IL-6 (10 ng/mL). Colonies were picked on day 7 of culture and transferred to RPMI liquid culture medium supplemented with 20% FBS, 1% PSG, SCF (40 ng/ml), GM-CSF (20 ng/ml), Flt3L (20 ng/ml), IL-3 (20 ng/ml), and IL-6 (20 ng/ml). To evaluate the sensitivity of *MLL-AF9* leukemic cells to HMAs in vivo, the leukemic cells were transplanted into lethally irradiated recipient mice, with support from CD45.1 mouse BM cells. Recipient mice were treated with DAC for 2 weeks, starting at 4 weeks after

 

transplantation. After the cessation of the treatment, the mice were euthanized for analysis.

## Culture of c-Kit-positive cells

The c-Kit-positive cells were bound to anti-Mouse CD117 Microbeads (Miltenyi Biotec) and purified twice using LS Columns (Miltenyi Biotec). Cells were cultured in SF-O3 supplemented with 0.2% BSA, 1% PSG, SCF (50 ng/mL), thrombopoietin (TPO; 50 ng/mL), IL-3 (50 ng/mL), and IL-6 (50 ng/mL) in the presence or absence of DAC.

## *TOPORS* knockout in umbilical cord blood CD34-positive cells

Cord blood cells was obtained from Japanese Red Cross Kanto-Koshinetsu Cord Blood Center with informed consent by participants. CD34-positive cells were expanded and washed twice with PBS, pelleted, and resuspended in 20 µl electroporation buffer P3 (Lonza). Recombinant S. pyogenes Cas9 (S.p. Cas9 Nuclease V3, IDT) was complexed with single guide RNA (sgRNA, synthesized at IDT) at a molar ratio of 1:2.5 for 10 min at 25 °C to form ribonucleoprotein (RNP) complexes. The sequence of sgRNA targeting human *TOPORS* was 5′-GTTTTCGCTGTGTACAGGAG-3′. The RNP duplex was gently added to the cells, and the suspension was transferred into a single 20 µL electroporation cuvette on a 16-well strip (P3 Primary Cell 96-well-Nucleofector Kit, Lonza). Electroporation was performed using the DZ-100 program on a 4D Nucleofector device (Lonza). After electroporation, the cells were immediately recovered in a pre-warmed medium and gently split into 24-well plates. One day after nucleofection, the medium was changed with subsequent medium changes performed every 2–3 days. The cells were cultured in Serum-Free Expansion Medium II (STEMCELL Technologies) supplemented with SCF (50 ng/mL), thrombopoietin (TPO; 50 ng/mL), IL-3 (50 ng/mL), and IL-6 (50 ng/mL).

## Western blotting and immunoprecipitation

Cells were washed twice with PBS and lysed in 2% SDS lysis buffer (10 mM Tris HCl, pH 8.0, 2% SDS). The DNA was cleaved using benzonase (Sigma-Aldrich). Lysates were then mixed with an equal volume of 2xSDS sample buffer (100 mM Tris-HCl, pH 6.8, 20% glycerol, 2% SDS, 0.04% BPB, and 10% 2-mercaptoethanol) and heated at 95 °C for 5 min for denaturation. Proteins in lysates were separated by SDS-PAGE, transferred to a PVDF membrane, and detected by western blotting using the following antibodies: anti-TOPORS (A302-179A, Bethyl Laboratories), anti-DNMT1 (#5032, Cell Signaling Technology), anti-UHRF1 (AB_399581, BD Biosciences), anti-SUMO1 (#4930, Cell Signaling Technology), anti-SUMO2/3 (ab3742, Abcam), anti-Ubiquitin (sc-8017, SANTA CRUZ), anti-UBE2D3 (ab176568, Abcam) and anti-β-actin antibody (#4967, Cell Signaling Technology). Protein bands were detected using an enhanced chemiluminescence reagent (Immobilon Western; Millipore). For immunoprecipitation, cells were lysed in denaturing buffer (20 mM Tris, pH7.5; 50 mM NaCl, 1 mM EDTA, 0.5% NP-40, 0.5% SDS, 0.5% sodium deoxycholate, 1 mM DTT) supplemented with protease inhibitors, followed by sonication. Cell lysates were cleared by centrifugation (16,000 × g, 5 min) and supernatants were incubated with anti-DNMT1 nanobody conjugated with agarose (DNMT1-trap agarose) (Chromotek) and for 1 h with constant agitation at 4 °C. After extensive washing of the beads, the targeted protein was eluted by boiling 2xSDS sample buffer for 5 min at 95 °C. A subcellular Protein Fractionation Kit for Cultured cells (Thermo Fisher Scientific) was used to prepare the fractionated samples.

## Mass spectrometry

WT and *TOPORS*-KO MDS-L cells were treated with 10 µM of DAC for 2 h each. The cells were lysed in 8 M urea-containing benzonase (Novagen). Samples were digested with Mass Spectrometry Grade Trypsin Gold (Promega). Ubiquitinated peptides were enriched using a PTMScan® HS Ubiquitin/SUMO Remnant Motif (K-ε-GG) Kit (Cell Signaling Technology). K-ε-GG-enriched peptides were analyzed using an Orbitrap Eclipse Tribrid mass spectrometer with a FAIMS Pro interface (Thermo Fisher Scientific) coupled to an UltiMate 3000 RSLCnano pump (Thermo Fisher Scientific). Data analysis was carried out based on a search against the UniProt human reference proteome database (UP000005640) using Proteome Discoverer 2.5 (Thermo). The abundance of ubiquitinated DNMT1 was normalized to the abundance of total DNMT1, which was measured by shotgun mass spectrometric analysis of WT and *TOPORS*-KO MDS-L cells.

## Patient samples and patient xenograft model

Patient samples were collected and preserved at the Komagome Hospital after obtaining written informed consent for use in the experiments. No financial compensation or material reward were provided for participating in this study. The StemSpan Leukemic Cell Culture Kit (STEMCELL Technologies, ST-09720) was used for in vitro culture. To establish PDX models, $1 \times 10^6$ patient (CH0680) cells were transplanted into NOG-W41/IL-3/GM-CSF-immunodeficient mice via tail vein injection without preconditioning. Two months after transplantation, chimerism of human CD45 cells in the BM was confirmed to be >90%, then $5 \times 10^6$ BM cells were transplanted into NOG/IL-3/GM-CSF mice irradiated at a dose of 1.5 Gy. After 3 weeks, the recipient mice were randomly divided into 4 groups and treated with either DAC and/or TAK-243 or vehicle for 2 weeks. After the cessation of the treatment, the mice were euthanized for analysis.

## RNA-seq data in patient samples

We selected 23 patients who received azacitidine treatment for myeloid malignancies from the Kyoto University Hematological Disease Biobank. All patient samples were collected and preserved after obtaining written informed consent for use in the experiments. No financial compensation or material reward were provided for participating in this study. The attending physicians judged the response to azacitidine as good ($n = 5$), partial ($n = 6$), or ineffective ($n = 12$). Sex and gender of participants was determined based on self-report but was not considered in this study. RNA (RIN value > 6.0) was extracted from pretreated tumor cells, prepped with the NEB Single-Cell/LowInput RNA Library Prep kit, and sequenced with Novaseq 6000 or DNB-SEQ. Mapping was performed using STAR-2.7.7 and read counts were calculated using Rsubread-1.32.4. We calculated the transcript per million (TPM) values for *TOPORS* and *DNMT1*.

## Statistical and reproducibility

Experimental data are presented as the mean ± standard error of the mean (SEM). Statistical analyses were performed using two-tailed Student's *t*-test or one-way ANOVA with Tukey–Kramer's post-hoc test. A significant difference was considered to exist when the *P*-value was less than 0.05. Statistical analyses were performed using GraphPad Prism (version 9). All in vitro experiments were performed in triplicate or more and repeated at least twice. We then used representative experimental data. No data were excluded from the analyses.

## Study approval

The present study was approved by the Ethics Committee of the Institute of Medical Science, University of Tokyo (Approval #2020-30-0917) and Tokyo Metropolitan Cancer and Infectious Diseases Center, Komagome Hospital (Approval #2203). RNA-seq of the clinical samples was approved by the Institutional Ethical Committee of Kyoto University (Approval #G-608).

## Reporting summary

Further information on research design is available in the Nature Portfolio Reporting Summary linked to this article.

## Data availability

The RNA sequence data were deposited in the DNA Data Bank of Japan under an accession number DRA015263. The mass spectrometry proteomics data were deposited to the ProteomeXchange Consortium via the jPOST repository with the dataset identifiers PXD044347 and PXD048275 [https://repository.jpostdb.org/preview/13665477056597fe513e282](access key: 6577). Source data are provided with this paper.

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

## Acknowledgements

The authors thank Dr. Hiroshi Matsuoka for providing the SKK-1 cell line; Dr. Satoshi Iwano and Atsushi Miyawaki for providing the Akaluc system; Dr. Stefan Weger for *TOPORS* plasmids; and Drs. Niels Maida, Carl Walkley, and Yasushi Saeki for their valuable advice. Supercomputing resource was provided by the Human Genome Center at the Institute of Medical Science, The University of Tokyo. This work was supported in part by Grants-in-Aid for Scientific Research (19H05653,19H05746 to AI, 19H05745 to HK) from the Japan Society for the Promotion of Science (JSPS), the Moonshot Project (21zf0127003h0001 to AI, JP22zf0127008 to HK) from the Japan Agency for Medical Research and Development (AMED), and the IMSUT Future Medical Development Fund. Satoshi Kaito was supported by JSPS scholarships (JP20J21190 and JP23KJ2189).

## Author contributions

Contribution: S.Kaito. and A.I. designed this study; S.Kaito. performed experiments, analyzed data, and actively wrote the manuscript; K.A., M.Oshima., A.T., M.M., S.Koide., Y.N.-T., H.K.-H., M.Oyama., T.Yogo., C.L., K.C.K., H.Kosako., and S.Y. performed experiments and analyzed data; K.Y. and Y.F. performed RNA sequencing; T.Yabushita., M.U. and A.H. provided the plasmids; K.T. and S.G. provided cell lines and transformed human leukemia cells; K.A. and K.H. provided the sgRNA library; N.D., Y.H., and H.H. collected human samples; R.I., and H.Koseki. generated mice; Y.N. and S.O. analyzed RNA data of patient samples; M.Y., A.Y., A.N., and M.N. discussed the results; A.I. guided and supervised the project, secured funding, and wrote the paper. All the authors have edited and approved the manuscript.

## Competing interests

K.H. is a co-founder of Dania Therapeutics, a consultant for Inthera Bioscience AG, and a scientific advisor for MetaboMed Inc. and Hannibal Innovation. All other authors declare no competing interests.

## Additional information

**Satoshi Kaito**[1,2], **Kazumasa Aoyama**[1,3], **Motohiko Oshima**[1], **Akiho Tsuchiya**[1], **Makiko Miyota**[1], **Masayuki Yamashita**[1,4], **Shuhei Koide**[1], **Yaeko Nakajima-Takagi**[1], **Hiroko Kozuka-Hata**[5], **Masaaki Oyama** ®[5], **Takao Yogo** ®[6], **Tomohiro Yabushita**[7], **Ryoji Ito** ®[8], **Masaya Ueno** ®[9], **Atsushi Hirao** ®[9], **Kaoru Tohyama**[10], **Chao Li**[11], **Kimihito Cojin Kawabata**[12], **Kiyoshi Yamaguchi**[13], **Yoichi Furukawa**[13], **Hidetaka Kosako** ®[14], **Akihide Yoshimi** ®[2], **Susumu Goyama** ®[15],

Yasuhito Nannya[16], Seishi Ogawa[17,18], Karl Agger[19], Kristian Helin [19,20], Satoshi Yamazaki[6,21], Haruhiko Koseki [22,23], Noriko Doki [24], Yuka Harada[25], Hironori Harada [24,26], Atsuya Nishiyama [27], Makoto Nakanishi [27] & Atsushi Iwama [1,28] ✉

[1]Division of Stem Cell and Molecular Medicine, Center for Stem Cell Biology and Regenerative Medicine, The Institute of Medical Science, The University of Tokyo, Tokyo, Japan. [2]Division of Cancer RNA Research, National Cancer Center Research Institute, Tokyo, Japan. [3]Division of Hygienic Chemistry, Faculty of Pharmacy, Keio University, Tokyo, Japan. [4]Division of Experimental Hematology, Department of Hematology, St. Jude Children's Research Hospital, Memphis, TN, USA. [5]Medical Proteomics Laboratory, The Institute of Medical Science, The University of Tokyo, Tokyo, Japan. [6]Division of Cell Regulation, Center for Experimental Medicine and Systems Biology, The Institute of Medical Science, The University of Tokyo, Tokyo, Japan. [7]Division of Cellular Therapy, The Institute of Medical Science, The University of Tokyo, Tokyo, Japan. [8]Central Institute for Experimental Animals, Yokohama, Kanagawa, Japan. [9]Cancer Research Institute, Kanazawa University, Kanazawa, Japan. [10]Department of Laboratory Medicine, Kawasaki Medical School, Okayama, Japan. [11]Department of Computational Biology and Medical Sciences, Graduate School of Frontier Sciences, The University of Tokyo, Tokyo, Japan. [12]Division of Clinical Precision Research, The Institute of Medical Science, The University of Tokyo, Tokyo, Japan. [13]Division of Clinical Genome Research, Advanced Clinical Research Center, Institute of Medical Science, The University of Tokyo, Tokyo, Japan. [14]Division of Cell Signaling, Fujii Memorial Institute of Medical Sciences, Institute of Advanced Medical Sciences, Tokushima University, Tokushima, Japan. [15]Division of Molecular Oncology, Department of Computational Biology and Medical Sciences, Graduate School of Frontier Sciences, The University of Tokyo, Tokyo, Japan. [16]Division of Hematopoietic Disease Control, Institute of Medical Science, The University of Tokyo, Tokyo, Japan. [17]Department of Pathology and Tumor Biology, Graduate School of Medicine, Kyoto University, Kyoto, Japan. [18]Institute for the Advanced Study of Human Biology (WPI-ASHBi), Kyoto University, Kyoto, Japan. [19]BRIC University of Copenhagen, Copenhagen, Denmark. [20]The Institute of Cancer Research (ICR), London, UK. [21]Division of Cell Engineering, Center for Stem Cell Biology and Regenerative Medicine, The Institute of Medical Science, The University of Tokyo, Tokyo, Japan. [22]Laboratory for Developmental Genetics, RIKEN Center for Integrative Medical Sciences, Yokohama, Japan. [23]Department of Molecular and Cellular Medicine, Graduate School of Medicine, Chiba University, Chiba, Japan. [24]Hematology Division, Tokyo Metropolitan Cancer and Infectious Diseases Center, Komagome Hospital, Tokyo, Japan. [25]Clinical Research Support Center, Tokyo Metropolitan Cancer and Infectious Diseases Center, Komagome Hospital, Tokyo, Japan. [26]Laboratory of Oncology, School of Life Sciences, Tokyo University of Pharmacy and Life Sciences, Tokyo, Japan. [27]Division of Cancer Cell Biology, Institute of Medical Science, University of Tokyo, Tokyo, Japan. [28]Laboratory of Cellular and Molecular Chemistry, Graduate School of Pharmaceutical Sciences, The University of Tokyo, Tokyo, Japan. ✉e-mail: 03aiwama@ims.u-tokyo.ac.jp

