## [Peer Review File · Nature Communications]

Inhibition of TOPORS ubiquitin ligase augments the efficacy of DNA hypomethylating agents through DNMT1 stabilizationREVIEWER COMMENTS

Reviewer #1 (Remarks to the Author); expert in AML:

The authors identified an ubiquitin/SUMO E3 ligase, TOPORS, implicated in the efficacy of DNA hypomethylating agents (HMAs). CRISPR-Cas9 screening revealed that knocking out TOPORS enhances the efficacy of HMAs in AML cells without affecting normal hematopoiesis, indicating TOPORS' involvement in HMA resistance. HMAs trap DNMT1 in DNA, forming DNA-DNMT1 crosslinks, and they propose that TOPORS is responsible for resolving these crosslinks through ubiquitination of SUMOylated DNMT1. They also showed that inhibitors targeting ubiquitination and SUMOylation synergize with HMAs by stabilizing DNMT1, offering a novel therapeutic strategy for HMA-resistant myeloid malignancies by interfering with the resolution of DNA-DNMT1 crosslinks. Overall, the manuscript is well written with generally supportive and rigorous data. The role of TOPORS in HMA resistance is novel and may reveal new therapeutic strategies. The notable concern of the study is related to the interpretation of their mechanistic findings. Specifically, it is not clear that stabilization of DNMT1 via ubiquitination is the mechanism by which TOPORS influences sensitivity to HMAs.

Major comments/suggestions:

- What are the protein levels of TOPORS in healthy versus AML cell lines and patient derived samples. The correlation at the RNA level (Figure 6F,G) is not as meaningful as examining protein expression, particularly that the authors propose that TOPORS regulates DNMT1 expression via post-translational modifications.
- TOPORS protein expression in cell lines should be correlated with HMA response.
- Figure 2. Deletion of TOPORS convincingly makes the cells more sensitive to HMA treatment. Does overexpression of TOPORS make sensitive cells resistant to HMA?
- Figure 3. What is the expression of TOPORS in MLL-AF9 vs control/normal HSPCs?
- Figure 5B. Expression of UBE2D3 protein in UBE2D3 KO cells should be shown.
- Figure 5F-G. It is paradoxical that TOPORS KO cells exhibit an increase in higher molecular weight DNMT1 protein, likely due to increased ubiquitination. Based on the proposed hypothesis, deletion of TOPORS, an E3 ligase, should result in diminished ubiquitination of DNMT1. This data is confusing.
- Figure 5F. The increase in DNMT1 protein expression in TOPORS KO cells relative to WT cells isn't significant.
- Figure 6D. As noted above, it is confusing that TOPORS KO cells exhibit increased ubiquitination and sumoylated of DNMT1. The total levels of DNMT1 do not appear different between KO and WT cells. Perhaps the authors should examine DNMT1 protein levels in nuclear vs cytosolic fractions.
- Figure 7. The combined effects of TAK-243 and DAC on the suppression of MDSL and MOLM13 cells is minor. These effects may be more prominent in patient derived samples. I would select AML samples that exhibit higher levels of TOPORS protein expression.

Minor comments/suggestions:

- Figure 1. It would be helpful to show a VENN diagram of the overlapping and distinct targets.
- The following statement (line 331) is not clear. I suggest rewording it. "DNMT1 underwent ubiquitination even in the absence of TOPORS (Fig. 6d), probably by RNF4, but accumulation of modified DNMT1 in TOPORS-KO cells indicated inefficient proteasomal degradation of DNMT1 in the absence of TOPORS."

Reviewer #2 (Remarks to the Author); expert in AML and CRISPR-Cas9 screening:

The authors present a comprehensive exploration of the therapeutic efficacy of DNA hypomethylating agents (HMAs) in myeloid malignancies, pinpointing TOPORS as a potential linchpin in HMA resistance mechanisms. The CRISPR-Cas9 screening, undertaken in two distinct cell lines, one of which models MDS, is an exemplary approach that enhances our grasp on the discussed mechanism. Further, the authors validate their findings both in vitro and in vivo, illustrating that a TOPORS knockout selectively impedes growth in the presence of HMA treatment. The deployment of Topors knockout (KO) mice provides valuable insights into TOPORS's role in

normal hematopoiesis – essentially indicating it as non-critical. The research's paramountcy is in its revelation of a novel resistance pathway in myeloid malignancies and the proposition of therapeutic interventions via TOPORS targeting.

Major Points:

1. To further fortify the manuscript's conclusions, the authors could consider presenting a dose-response curve of DAC in both WT and TOPORS knockout cells. This additional data could provide a deeper understanding of the drug's specificity and efficacy in the context of TOPORS-mediated HMA resistance.
2. While the employment of CRISPR-Cas9 screening is laudable, an explicit presentation of associated p-values, particularly for TOPORS, would bolster the study's statistical rigor and credibility.
3. The authors provide a detailed account of the impact on human cord blood with MLL-AF9. It would be pertinent to delineate any conceivable off-target or non-specific effects on normal CD34+ HSPCs, enhancing the study's therapeutic implications.
4. For a more translational applicability, it would be advantageous to validate the findings in primary patient samples in vitro and in vivo, especially when assessing the effect of the combination therapy.
5. A recommendation for the authors would be to undertake synergy studies to bolster the narrative around the combined effect of DAC with TAK-243 and TAK-981. Employing a synergy score, like BLISS, could quantify the therapeutic efficacy of the combination.

Minor Points:

1. To uphold the clarity and credibility of the study, addressing the mislabeling of the y-axis in Figure 2B and similar figures is advised.

Reviewer #3 (Remarks to the Author); expert in DNA damage and post-translational modifications:

In this manuscript, Kaito et al., prove that TOPORS have a role in the resolution on DNMT1-DNA crosslinks after exposure to HMAs, explore its potential as a target for leukemia treatment. The manuscript is excellently written, and the data is solid and of high quality. However, this reviewer feels that some claims/statements are not sufficiently supported by the data and either extra work to support them or toned down.

Detailed hereunder:

Major concerns.

- In figure 5a, Wt by not RD mutant of TOPORS rescue the sensitivity to DAC. However, RD deletion might affect the stability of the protein. Analysis by immunoblotting (or MS proteomics in case the antibody epitope falls into the RD domain) against TOPORS should be provided to exclude the possibility that the difference is due to the lack of the catalytic activity. RING domains also promote protein-protein interactions, more "conservative" mutations should be performed, like C to S in the Zn²⁺ cysteine.

- UBE2D3 appears as a hit for DAC sensitivity in MDS-L cells but not in MOLM-13 cells (Figure 1). With the current data, linking UBE2D3 sensitivity to DAC with being the E2 associated with TOPORS and responsible of DAC sensitivity, is an overstatement. Mainly because UBE2D3 is also associated with many other E3 enzymes involved in DNA repair that might be required upon collision of the replication/transcription machinery with crosslinked DNMT1. To support this conclusion, a double mutant TOPORS-KO UBE2D3-KO should be included in figure 2B and observe epistasis. Removing this data/statement would not affect either the message of the paper.

Minor points

- in line 137, there is a call for figure 3 which is not correct. Should be Extended figure 3
- Being Nature communs a generalistic journal authors should explain why is relevant in lines 167 – 170 he difference between NOG mice and NOG mice expressing human IL-3 and GM-CSF.
- Legend is missing in Figure 3f, is intuitive that red corresponds to WT and Blue to Topors -/-
- IN lines 240 – 245 The authors explain that differences in gene expression correlate with differences in cell cycle upon DAC treatment. However, this is also the case in the absence of DAC, so clearly other genes are affected by TOPORS deficiency. However this is not relevant for the message and could be omitted from the main figures (even from the manuscript).

Other comments,

The conclusions of this work are additionally supported by recent publications on the use of SUMOylation inhibitors in combination with HMAs to treat Leukemias (Kroonen et al. 2023, Leukemia PMID: 36792656; Gabellier et al. 2023 Haematologica PMID: 37608777), which should be cited. This manuscript adds further mechanistical insight and proposes TOPORS as a STUBL responsible of this process, this is very interesting data that could be further supported increasing the outreach of this manuscript.

RNF4 knockdown produces an accumulation of SUMO2/3 conjugates but not SUMO1 (Tatham et al. 2008. Nat Cell Bio), TOPORS has a SUMO1 interacting region apart from other SIMs, but has affinity for all SUMO isoforms and SUMO2/3 chains as well (González-Prieto et al. 2021 Cell Reports). There is the possibility that TOPORS is a SUMO1 STUBL. It would be worth testing by immunoblotting whether TOPORS-KO cells have an accumulation of SUMO1, or SUMO2/3 o conjugates or maybe both compared to their wild type counterparts. Alternatively, immunoprecipitation of DNMT1 after DAC treatment in denaturing conditions and blotting against SUMO1 or SUMO2/3 to see with which SUMO type is DNMT1 modified upon DAC treatment in Leukemia cells.

To Reviewer 1:

The authors identified an ubiquitin/SUMO E3 ligase, TOPORS, implicated in the efficacy of DNA hypomethylating agents (HMAs). CRISPR-Cas9 screening revealed that knocking out TOPORS enhances the efficacy of HMAs in AML cells without affecting normal hematopoiesis, indicating TOPORS' involvement in HMA resistance. HMAs trap DNMT1 in DNA, forming DNA-DNMT1 crosslinks, and they propose that TOPORS is responsible for resolving these crosslinks through ubiquitination of SUMOylated DNMT1. They also showed that inhibitors targeting ubiquitination and SUMOylation synergize with HMAs by stabilizing DNMT1, offering a novel therapeutic strategy for HMA-resistant myeloid malignancies by interfering with the resolution of DNA-DNMT1 crosslinks. Overall, the manuscript is well written with generally supportive and rigorous data. The role of TOPORS in HMA resistance is novel and may reveal new therapeutic strategies. The notable concern of the study is related to the interpretation of their mechanistic findings. Specifically, it is not clear that stabilization of DNMT1 via ubiquitination is the mechanism by which TOPORS influences sensitivity to HMAs.

Response: Thank you for your valuable comments and suggestions. Based on the reviewers' comments, we have performed additional experiments, and the manuscript has been greatly improved. As indicated in the following responses, we have taken all these comments and suggestions into account when preparing the revised version of our manuscript. All the revisions are highlighted in red.

Major comments/suggestions:

- What are the protein levels of TOPORS in healthy versus AML cell lines and patient derived samples. The correlation at the RNA level (Figure 6F,G) is not as meaningful as examining protein expression, particularly that the authors propose that TOPORS regulates DNMT1 expression via post-translational modifications.
- TOPORS protein expression in cell lines should be correlated with HMA response.

Response: Thank you for the suggestion. First, we examined TOPORS protein levels by western blotting using eight cell lines (MOLM-13, MDS-L, SKK-1, SKM-1, HL-60, THP-1, K562, and HeLa) and umbilical cord blood (CB)-derived CD34-positive cells as healthy controls (Extended Data Fig 5c). Unfortunately, evaluation of protein levels in

patient samples using western blotting has not been successful, mainly because of the difficulty in obtaining sufficient cells. The sensitivity of each cell line to DAC was evaluated by calculating the absolute IC50 based on the results of MTS assays (Extended Data Fig 5b). It may be difficult to explain resistance to DAC solely by TOPORS protein levels, because various factors may influence resistance to DAC. However, there was a trend toward a positive correlation between TOPORS protein levels and resistance to DAC (absolute IC50 values), although the correlation was not statistically significant ($P=0.075$) (Extended Data Fig 5d). We presented these data in Extended Data Fig. 5 and described these points on page 10, lines 183-190.

We also assessed *TOPORS* mRNA expression levels using RT-PCR in each cell line and in cord blood (CB)-derived CD34-positive cells. However, they were poorly correlated with protein levels, as shown below. As the reviewer suggested, TOPORS protein levels, rather than mRNA expression levels, may affect resistance to hypomethylating agents. Therefore, in the revised manuscript, figures showing the correlation between mRNA levels in patient samples and therapeutic responses to HMA have been excluded from the main figure (Figure 6f and g) and moved to Extended Data Fig. 11. We have deleted this statement from the original abstract.

- Figure 2. Deletion of TOPORS convincingly makes the cells more sensitive to HMA treatment. Does overexpression of TOPORS make sensitive cells resistant to HMA?

Response: Thank you for your comment regarding the impact of TOPORS overexpression on sensitivity to HMA. We evaluated the effect of TOPORS overexpression on the

sensitivity to DAC using MOLM-13 and MDS-L cells retrovirally transduced with *TOPORS*. Overall, changes in DAC resistance by *TOPORS* overexpression were not evident in either cell line as shown below. Overexpression of exogenous *TOPORS* was hardly detected by western blotting (data not shown), although it was statistically significant at mRNA levels by RT-qPCR. It is possible that an excess of *TOPORS* protein is toxic to cells and precisely regulated by proteasomal degradation.

- Figure 3. What is the expression of *TOPORS* in *MLL-AF9* vs control/normal HSPCs?

Response: We checked the expression level of *Topors* in murine *MLL-AF9* leukemia cells, but the *Topors* protein was not clearly detected by western blotting using currently available antibodies. Instead, RT-PCR revealed that *Topors* mRNA levels were moderately but significantly higher in leukemic cells than in normal c-Kit-positive HSPCs. We showed these data in Extended data Fig 6b and described this point on page 12, lines 242-245.

- Figure 5B. Expression of *UBE2D3* protein in *UBE2D3* KO cells should be shown.

Response: Thank you for your comment regarding the *UBE2D3* knockout. Western blotting confirmed that the protein levels of *UBE2D3* were decreased in MDS-L-knockout cells (Extended Data Fig. 8d in the revised version). Following the suggestion

of Reviewer 3, we moved the UBE2D3 data to Extended Data Fig 8d.

Figure 5F-G. It is paradoxical that TOPORS KO cells exhibit an increase in higher molecular weight DNMT1 protein, likely due to increased ubiquitination. Based on the proposed hypothesis, deletion of TOPORS, an E3 ligase, should result in diminished ubiquitination of DNMT1. This data is confusing.

Response: Thank you for your comment regarding our western blot data, particularly the band shift and expression levels of DNMT1 (Figure 5e and f in the revised version). Based on our hypothesis that TOPORS functions as a ubiquitin ligase for SUMOylated DNMT1, SUMOylated DNMT1 was expected to accumulate in *TOPORS*-KO cells. Immunoprecipitation of DNMT1 8 h after exposure to high concentrations of DAC clearly confirmed the accumulation of SUMOylated DNMT1 as high-molecular-weight DNMT1 (figure 6d in the original version, moved to Extended Data Fig 9b in this revised version), which disappeared upon addition of the SUMOylation inhibitor ML-792 (Figure 5f). We described this point on page 17, lines 357-359.

In the revised manuscript, we have also conducted immunoprecipitation 2 hours after exposure to high concentrations of DAC under the same conditions as the mass spectrometry analysis in Figure 6c. As expected, SUMOylated DNMT1 accumulated in *TOPORS*-KO cells (Figure 6d). In addition, the ubiquitination levels of SUMOylated DNMT1 decreased in *TOPORS*-KO cells (Figure 6d). These data were consistent with the mass spectrometry results (Figure 6c). Furthermore, we normalized the mass spectrometry data of DNMT1 ubiquitination to the actual amount of DNMT1 measured by shotgun mass spectrometric analysis of WT and *TOPORS* KO MDS-L cells (Figure 6c) and confirmed that DNMT1 ubiquitination was significantly reduced in *TOPORS*-KO cells.

As the reviewer pointed out, DNMT1 undergoes ubiquitination even in the absence of TOPORS. We speculate that SUMOylated DNMT1 is incompletely ubiquitinated, probably by RNF4, in the absence of TOPORS, as detected by mass spectrometry, and undergoes proteasomal degradation much less efficiently than in WT cells. Thus, SUMOylated DNMT1, which accumulates as high-molecular-weight DNMT1, is also ubiquitinated at low levels.

We have presented these data in Fig. 6c and d and described this point on pages 18-

19, lines 379-383 and 384-395.

- Figure 5F. The increase in DNMT1 protein expression in TOPORS KO cells relative to WT cells isn't significant.

Response: Thank you for your comment regarding Figure 5f (Figure 5e in the revised version). In this study, we performed western blotting after exposure to a very low concentration (12.5 nM) of DAC, as our screening was performed at this concentration, and MDS-L showed a prominent difference in the growth rate between wild-type and *TOPORS*-KO cells. As the reviewer has pointed out, it was difficult to observe a clear difference in the degradation rate of DNMT1 (main band) by western blotting at this low concentration of DAC. In contrast, the slowly migrating SUMOylated sub-bands were enhanced after DAC exposure in *TOPORS*-KO cells, even at low DAC concentrations. We have toned down our statement to avoid overstatement on page 16, lines 331-340.

To obtain clearer differences between WT and *TOPORS*-KO cells, we next exposed cells with high concentrations of DAC (10 μ M). We did see a difference in the degradation rate of DNMT1 and prominently enhanced slowly migrated DNMT1 bands after a short exposure to high concentrations (10 μ M) of DAC (Figure 5f). Correspondingly, there were clear differences in DNMT1 degradation between MOLM-13 and murine leukemia cells exposed to high concentrations of DAC (Figure 5g-i). We have described these points on pages 16-17, lines 340-345.

Figure 6D. As noted above, it is confusing that TOPORS KO cells exhibit increased ubiquitination and sumoylated of DNMT1. The total levels of DNMT1 do not appear different between KO and WT cells. Perhaps the authors should examine DNMT1 protein levels in nuclear vs cytosolic fractions.

Response: Thank you for your comments on the post-translational modification of DNMT1 in *TOPORS*-KO cells. As mentioned above, TOPORS acts as a ubiquitin ligase for SUMOylated DNMT1, suggesting that SUMOylated DNMT1 accumulated in *TOPORS*-KO cells. In accordance with the reviewer's comment, we assessed the DNMT1 protein levels using fractionated samples. As expected, the slow migration of DNMT1, which represents SUMOylated DNMT1, was more evident in *TOPORS*-KO cells in the

chromatin-bound fraction (Extended Data Fig. 9c). This may represent DNA-trapped SUMOylated DNMT1 and supports our hypothesis. We described this point on page 17, lines 357-363.

- Figure 7. The combined effects of TAK-243 and DAC on the suppression of MDSL and MOLM13 cells is minor. These effects may be more prominent in patient derived samples. I would select AML samples that exhibit higher levels of TOPORS protein expression.

Response: Thank you for the comment. As the reviewer pointed out, we agree that it is important to test primary patient samples to demonstrate the significance of this combination therapy with DAC and TAK-243.

In the revised manuscript, we first established an *in vitro* culture system and confirmed that DAC and TAK-243 exhibited synergistic effects at certain concentrations in primary AML samples (Figure 8a and b). We then established a patient-derived xenograft model of AML and showed that the combination of DAC and TAK-243 is effective in this model as well (Figure 8c and d). These results suggest that a combination of HMAs and TAK-243 may be useful in clinical practice. We have added the data to Figure 8 and described these data on pages 20-21, lines 432-449.

Minor comments/suggestions:

- Figure 1. It would be helpful to show a VENN diagram of the overlapping and distinct targets.

Response: Thank you for the suggestion. We have added Venn diagrams, as shown in Figure 1g.

- The following statement (line 331) is not clear. I suggest rewording it. “DNMT1 underwent ubiquitination even in the absence of TOPORS (Fig. 6d), probably by RNF4, but accumulation of modified DNMT1 in TOPORS-KO cells indicated inefficient proteasomal degradation of DNMT1 in the absence of TOPORS.”

Response: Thank you for your comment. We have made the following corrections (pages

18-19, lines 391-395). “DNMT1 was ubiquitinated at lower levels even in the absence of TOPORS (Fig. 6c, d), probably by RNF4. However, considerable accumulation of SUMOylated DNMT1 in *TOPORS*-KO cells upon HMA exposure clearly indicated inefficient proteasomal degradation of DNMT1 in the absence of TOPORS.”

To Reviewer 2:

The authors present a comprehensive exploration of the therapeutic efficacy of DNA hypomethylating agents (HMAs) in myeloid malignancies, pinpointing TOPORS as a potential linchpin in HMA resistance mechanisms. The CRISPR-Cas9 screening, undertaken in two distinct cell lines, one of which models MDS, is an exemplary approach that enhances our grasp on the discussed mechanism. Further, the authors validate their findings both in vitro and in vivo, illustrating that a TOPORS knockout selectively impedes growth in the presence of HMA treatment. The deployment of Topors knockout (KO) mice provides valuable insights into TOPORS's role in normal hematopoiesis – essentially indicating it as non-critical. The research's paramountcy is in its revelation of a novel resistance pathway in myeloid malignancies and the proposition of therapeutic interventions via TOPORS targeting.

Response: Thank you for your encouraging comments and suggestions. Based on the reviewers' comments, we have performed additional experiments, and the manuscript has been greatly improved. As indicated in the following responses, we have taken all these comments and suggestions into account when preparing the revised version of our manuscript. All the revisions are highlighted in red.

Major Points:

1. To further fortify the manuscript's conclusions, the authors could consider presenting a dose-response curve of DAC in both WT and TOPORS knockout cells. This additional data could provide a deeper understanding of the drug's specificity and efficacy in the context of TOPORS-mediated HMA resistance.

Response: Thank you for the suggestion. In this revised manuscript, we have presented dose-response curves of WT and *TOPORS*-KO MOLM-13 and MDS-L cells to DAC (Extended Fig Data Fig. 5a). Sensitivity to DAC was clearly different between the WT and *TOPORS*-KO cells at various DAC concentrations. We have described this on page 10, lines 183-186.

2. While the employment of CRISPR-Cas9 screening is laudable, an explicit presentation

of associated p-values, particularly for TOPORS, would bolster the study's statistical rigor and credibility.

Response: Thank you for the comment. *P* values are shown in figure 1f.

3. The authors provide a detailed account of the impact on human cord blood with MLL-AF9. It would be pertinent to delineate any conceivable off-target or non-specific effects on normal CD34+ HSPCs, enhancing the study's therapeutic implications.

Response: Thank you for your comment on the effect of *TOPORS* knockout on normal hematopoiesis. We agree with this point and performed additional experiments using human CD34-positive HSPCs. We introduced sgRNAs for *TOPORS* and Cas9 proteins into human CD34-positive HSPCs by electroporation and obtained nearly 90% gene editing efficiency. *TOPORS*-KO HSPCs did not show any significant changes in cell proliferation. These results indicate that the off-target effects of *TOPORS* inhibition were negligible. We present these data in Extended Data Fig. 4e and describe this point on pages 9-10, lines 175-179.

4. For a more translational applicability, it would be advantageous to validate the findings in primary patient samples in vitro and in vivo, especially when assessing the effect of the combination therapy.

Response: Thank you for your constructive suggestion. We agree that we must demonstrate the significance of this combination therapy in primary patient samples. In the revised manuscript, we established an *in vitro* culture system and confirmed that DAC and TAK-243 exhibited synergistic effects at certain concentrations in primary AML samples, although each sample showed a different response to these drugs (Figure 8a and b). We then established a patient-derived xenograft model of AML and showed that the combination of DAC and TAK-243 is effective in this model as well (Figure 8c and d). These results suggest that a combination of HMAs and TAK-243 may be useful in clinical practice. We presented these data in Figure 8 and described these points on pages 20-21, lines 432-449.

5. A recommendation for the authors would be to undertake synergy studies to bolster the narrative around the combined effect of DAC with TAK-243 and TAK-981. Employing a synergy score, like BLISS, could quantify the therapeutic efficacy of the combination.

Response: Thank you for the suggestion. In the revised manuscript, we have evaluated the synergies between HMAs and TAK-243 or TAK-981 using MOLM-13 and MDS-L. Although the combinations were not always synergistic or antagonistic at some concentrations, as shown in the original study, the combinations at lower concentrations showed synergistic effects. We have added the following figure as Extended Data Fig. 12c and described this point on page 20, lines 418-422.

Minor Points:

1. To uphold the clarity and credibility of the study, addressing the mislabeling of the y-axis in Figure 2B and similar figures is advised.

Response: Thank you for your comment. We have corrected them.

To Reviewer 3:

In this manuscript, Kaito et al., prove that TOPORS have a role in the resolution on DNMT1-DNA crosslinks after exposure to HMAs, explore its potential as a target for leukemia treatment. The manuscript is excellently written, and the data is solid and of high quality. However, this reviewer feels that some claims/statements are not sufficiently supported by the data and either extra work to support them or toned down.

Detailed hereunder:

Response: Thank you for the comment. Based on the reviewers' comments, we have performed additional experiments, and the manuscript has been greatly improved. As indicated in the following responses, we have taken all these comments and suggestions into account when preparing the revised version of our manuscript. All the revisions are highlighted in red.

Major concerns.

- In figure 5a, Wt by not RD mutant of TOPORS rescue the sensitivity to DAC. However, RD deletion might affect the stability of the protein. Analysis by immunoblotting (or MS proteomics in case the antibody epitope falls into the RD domain) against TOPORS should be provided to exclude the possibility that the difference is due to the lack of the catalytic activity. RING domains also promote protein-protein interactions, more "conservative" mutations should be performed, like C to S in the Zn²⁺ cysteine.

Response: Thank you for your comment regarding the rescue experiment. In a previous report, it was reported that a C-to-A mutation at C1 and C2 in the RING finger domain reduced ubiquitination activity (Front Cell Dev Biol. 2020;8:39). Therefore, we created C1 and C1C2 double mutants (Extended Data Fig. 8a) and performed addback experiments on *TOPORS*-KO MDS-L cells. The addback of these mutants was significantly less effective than WT TOPORS in canceling DAC resistance (Extended Data Fig. 8b). The C1 and C1C2 mutants showed phenotypes that were intermediate between those of the WT and RD mutants, possibly due to incomplete inactivation of catalytic activity (Extended Data Fig. 8b). These results further support the hypothesis

that the RING finger domain is responsible for DAC resistance.

We also evaluated the protein levels of TOPORS and TOPORS mutants by western blotting and found that the WT, C1 mutant, and C1C2 mutant proteins were hardly detected, whereas the RD mutant protein was readily detected (Extended Data Fig. 8c). We confirmed that the WT and all the mutants were overexpressed at the mRNA level, with the RD mutant being much more highly expressed than the others (data not shown). This was probably due to the proteasomal degradation of TOPORS by self-ubiquitination. Mass spectrometry revealed the ubiquitination of TOPORS at K249 (data not shown), which is considered a candidate site for self-ubiquitination. The RD mutant lacking the RING finger domain may escape self-ubiquitination due to a conformational change. Since the RD mutant, which is much more stable than WT TOPORS, scarcely canceled DAC resistance, and the RING finger point mutants were less effective in canceling DAC resistance, it would be reasonable to conclude that the RING finger is the responsible site. We have shown these data in Extended Data Fig. 8a-c and described them on pages 14-15, lines 285-298.

- UBE2D3 appears as a hit for DAC sensitivity in MDS-L cells but not in MOLM-13 cells (Figure 1). With the current data, linking UBE2D3 sensitivity to DAC with being the E2 associated with TOPORS and responsible of DAC sensitivity, is an overstatement. Mainly because UBE2D3 is also associated with many other E3 enzymes involved in DNA repair that might be required upon collision of the replication/transcription machinery with crosslinked DNMT1. To support this conclusion, a double mutant TOPORS-KO UBE2D3-KO should be included in figure 2B and observe epistasis. Removing this data/statement would not affect either the message of the paper.

Response: Thank you for your comment regarding UBE2D3. We agree with the reviewer's comment. We excluded the data from the main figure and moved it to Extended Data Fig. 8d with the new western blot data and deleted the original statement, linking UBE2D3 sensitivity to DAC with E2 associated with TOPORS and responsible for DAC sensitivity to avoid overstatement.

Minor points

- in line 137, there is a call for figure 3 which is not correct. Should be Extended figure 3

Response: Thank you for your helpful comments. We edited this mistake.

- Being Nature communs a generalistic journal authors should explain why is relevant in lines 167 – 170 the difference between NOG mice and NOG mice expressing human IL-3 and GM-CSF.

Response: Thank you for the comment. We have explained this as follows (page 10, lines 199-200). “Because MDS-L is an IL-3-dependent cell line, we used NOG mice expressing human IL-3 and GM-CSF for these xenograft experiments.”

- Legend is missing in Figure 3f, is intuitive that red corresponds to WT and Blue to Topors -/-

Response: Thank you for the comment. We have added the legend that red corresponds to WT and Blue to Topors -/-.

- IN lines 240 – 245 The authors explain that differences in gene expression correlate with differences in cell cycle upon DAC treatment. However, this is also the case in the absence of DAC, so clearly other genes are affected by TOPORS deficiency. However this is not relevant for the message and could be omitted from the main figures (even from the manuscript).

Response: Thank you for your constructive suggestion. Certainly, many gene expression was affected by *TOPORS*-KO even in the absence of DAC, and these data are unlikely to be relevant to the main message. We moved the RNA-seq data from the main figure to Extended Data Fig. 7c and deleted the GSEA plots and corresponding text from the main text.

Other comments,

The conclusions of this work are additionally supported by recent publications on the use of SUMOylation inhibitors in combination with HMAs to treat Leukemias (Kroonen et

al. 2023, Leukemia PMID: 36792656; Gabellier et al. 2023 Haematologica PMID: 37608777), which should be cited. This manuscript adds further mechanistical insight and proposes TOPORS as a STUBL responsible of this process, this is very interesting data that could be further supported increasing the outreach of this manuscript.

Response: Thank you for providing us with literature on the synergistic effects of HMAs and the SUMOylation inhibitor, TAK-981. In the revised version, we have cited and discussed these reports on page 24, lines 500-504.

RNF4 knockdown produces an accumulation of SUMO2/3 conjugates but not SUMO1 (Tatham et al. 2008. Nat Cell Bio), TOPORS has a SUMO1 interacting region apart from other SIMs, but has affinity for all SUMO isoforms and SUMO2/3 chains as well (González-Prieto et al. 2021 Cell Reports). There is the possibility that TOPORS is a SUMO1 STUBL. It would be worth testing by immunoblotting whether TOPORS-KO cells have an accumulation of SUMO1, or SUMO2/3 conjugates or maybe both compared to their wild type counterparts. Alternatively, immunoprecipitation of DNMT1 after DAC treatment in denaturing conditions and blotting against SUMO1 or SUMO2/3 to see with which SUMO type is DNMT1 modified upon DAC treatment in Leukemia cells.

Response: Thank you for your comments regarding possible differences in TOPORS affinity for SUMO1 and SUMO2/3. González-Prieto et al. showed that TOPORS is a binder with high affinity for SUMO1, SUMO2, and SUMO2 trimers using mass spectrometry (Cell Rep. 2021 Jan 26;34(4):108691). These results are consistent with our hypothesis that TOPORS functions as a STUBL in SUMOylated DNMT1. As shown in Figure 6d and Extended Data Fig. 9b, DNMT1 was highly modified by SUMO1 and SUMO2/3 upon DAC treatment. No clear differences in the abundance were observed. In the revised version, we have cited and discussed the report by González-Prieto et al. on page 23, lines 481-484.

REVIEWERS' COMMENTS

Reviewer #1 (Remarks to the Author):

The authors have satisfactorily addressed my concerns.

Reviewer #2 (Remarks to the Author):

The authors have thoughtfully and thoroughly addressed the concerns raised by this reviewer, resulting in substantial enhancements to the manuscript.

Reviewer #3 (Remarks to the Author):

The authors have successfully addressed all my concerns. The new data has improved the article. I would recommend acceptance and publication in Nature Communications.